# From Tables to Time:
# Extending TabPFN-v2 to time-series forecasting

**Shi Bin Hoo**                                    *hoos@tf.uni-freiburg.de*
*University of Freiburg*
*Prior Labs*

**Samuel Müller**[*]                               *sammuller@meta.com*
*University of Freiburg*
*Meta*

**David Salinas**                                  *david.salinas@tue.ellis.eu*
*ELLIS Institute Tübingen*
*University of Freiburg*

**Frank Hutter**                                   *frank@priorlabs.ai*
*Prior Labs*
*ELLIS Institute Tübingen*
*University of Freiburg*

**Reviewed on OpenReview:** *https://openreview.net/forum?id=KIkQj8VOUY*

## Abstract

Recent progress in foundation models has enabled strong zero-shot performance for time-series forecasting. In this work, we show that these capabilities also emerge from tabular foundation models. We introduce TabPFN-TS, a simple method that treats forecasting as a tabular regression problem by combining lightweight temporal featurization with the pretrained TabPFN-v2. This formulation requires no time-series–specific pretraining and naturally supports both univariate and covariate-informed forecasting. Despite its compact size (11M parameters), TabPFN-TS achieves state-of-the-art performance on covariate-informed forecasting and competitive accuracy on univariate forecasting across the GIFT-Eval and `fev-bench` benchmarks. We further provide controlled analyses examining how the model interprets temporal structure, how featurization choices affect accuracy, and how forecasts change under alternative tabular backbones. Together, our results demonstrate that tabular foundation models—when paired with suitable temporal features—offer an efficient and versatile alternative for forecasting, bridging tabular and time-series learning within a unified framework.

## 1 Introduction

Recent advances in foundation models have reshaped time-series forecasting by enabling zero-shot prediction across a wide variety of domains. Large pretrained architectures can now generalize to unseen series and horizons without task-specific fine-tuning, setting a new paradigm for forecasting research. Comprehensive benchmarks (Aksu et al., 2024; Shchur et al., 2025) have further standardized evaluation, enabling fair and reproducible comparison.

While most foundation models focus on univariate forecasting, real-world systems are often influenced by additional factors. Incorporating such covariates—e.g., weather, control inputs, or economic indicators—can make forecasts far more informative and actionable. Yet leveraging covariates remains challenging for

---

[*]Work conducted while at the University of Freiburg.

current time-series foundation models, which are typically designed around purely sequential inputs. A unified and data-efficient framework capable of handling both univariate and covariate-informed forecasting would therefore be highly desirable.

In parallel, foundation models for *tabular learning* have made remarkable progress. Recent tabular models exhibit strong zero-shot predictive performance across diverse supervised tasks (Erickson et al., 2025). Among them, TabPFN-v2 (Hollmann et al., 2025) is notable as the first tabular foundation model, supporting many tasks, including classification, regression, outlier detection, density estimation, synthetic data generation, embeddings that are useful for downstream tasks, and fine-tunability. This raises a natural question: *Can a tabular foundation model also perform probabilistic time-series forecasting if the temporal structure is expressed through features?* If so, tabular models would offer a unified representation for both tabular and time-series forecasting, sidestepping the need for specialized sequence architectures.

In this work, we propose to treat time-series forecasting as a *tabular regression problem*. We introduce TabPFN-TS, a simple method that combines lightweight temporal featurization with the pretrained TabPFN-v2, enabling zero-shot forecasting without any time-series–specific pretraining or fine-tuning. Each time step is represented as a row of temporal features (and optional covariates) paired with its observed value. Forecasting then reduces to predicting future rows—whose temporal features are known in advance—using TabPFN-v2 as a tabular regressor. This formulation allows all future points within the prediction horizon to be predicted in a single forward pass.

Despite its simplicity and compact size (11M parameters), TabPFN-TS achieves state-of-the-art performance on covariate-informed forecasting and competitive accuracy on univariate benchmark. Our analyses further show how the model interprets temporal structure, how featurization affects accuracy, and how forecasts change under different tabular backbones.

Our main contributions are:

1. **Forecasting as Tabular Regression.** We introduce a formulation that enables a pretrained tabular foundation model (TabPFN-v2) to perform both univariate and covariate-informed forecasting in a zero-shot manner.

2. **Lightweight Temporal Featurization.** We design a compact feature scheme that encodes time progression, multi-scale seasonality, and covariates, allowing tabular models to act on temporal data effectively.

3. **Strong Empirical Performance.** TabPFN-TS achieves state-of-the-art performance on covariate-aware forecasting and competitive accuracy on univariate forecasting across GIFT-Eval and `fev-bench`.

4. **Mechanistic and Ablation Studies.** We provide analyses explaining how TabPFN-TS exploits temporal structure, how featurization choices affect performance, and how forecasting quality varies across tabular foundation model backbones.

Taken together, these results demonstrate that tabular foundation models, paired with suitable temporal features, form a simple, efficient, and extensible alternative for general-purpose forecasting, bridging tabular and time-series learning within a unified framework.

## 2 Background and Related Work

**time-series forecasting.** The goal is to predict future observations of a sequence based on its historical context and, optionally, a set of covariates—external variables that provide additional information about the system's dynamics (e.g. holidays, known prices, or control inputs). Incorporating such signals can improve forecast accuracy by capturing influences beyond the intrinsic temporal patterns of the target series. Formally, the task can be expressed as modeling the conditional distribution

$$P(y_{C+1:H} \mid y_{1:C}, Z_{1:C+H}),$$

where $y_{1:C} = [y_1, \ldots, y_C]$ denotes the observed history of the target series, $y_{C+1:H} = [y_{C+1}, \ldots, y_H]$ are the future values to be predicted, and $Z_{1:C+H} = [\mathbf{z}_1, \ldots, \mathbf{z}_{C+H}]$ represents the associated covariates. When no covariates are provided, the task reduces to *univariate forecasting*; otherwise, it corresponds to *covariate-informed forecasting*.

**Time Series Foundation Models.** Time series foundation models (TSFMs) aim to provide zero-shot forecasting across domains by pretraining (potentially) large sequence models on broad collections of real-world time series. Early models such as Chronos (Ansari et al., 2024) and TimesFM (Das et al., 2024) adopt transformer architectures trained at scale for univariate forecasting. Chronos-Bolt improves efficiency beyond Chronos through patch-based encoding, while Toto (Cohen et al., 2024) targets multivariate forecasting with proportional factorized attention and a Student-$t$ mixture output layer. TiRex (Auer et al., 2025b) leverages an xLSTM backbone with strong in-context learning, and Sundial (Liu et al., 2025) introduces a flow-matching objective for flexible probabilistic forecasting.

Several TSFMs incorporate covariates. ChronosX (Arango et al., 2025) injects exogenous variables through modular adapters; and COSMIC (Auer et al., 2025a) augments forecasting with in-context covariate conditioning.[1]

**TabPFN-v2.** TabPFN (Hollmann et al., 2023) is a foundation model pretrained to perform tabular prediction directly from examples. Instead of being optimized on a single dataset, it is pretrained on millions of synthetic regression and classification tasks. Each task consists of input–output pairs $(X, y)$ as the context and query points $X'$ for which the model must predict corresponding targets $y'$. Through this large-scale pretraining, TabPFN learns general patterns of how features relate to targets that transfer across unseen problems.

At inference, the model receives a table of feature-target pairs $(X, y)$ as context and outputs the posterior predictive distribution $p(y' \mid X, y, x')$ for new inputs $X'$. TabPFN represents this distribution as a *Riemann distribution*—a discretized probability density over possible target values—and is trained directly on this probabilistic representation. Having the model output a distribution allows to handle downstream cases which requires a distribution, for instance to account for the model uncertainty.

TabPFN-v2 (Hollmann et al., 2025) extends its predecessor TabPFN with architectural improvements, a richer synthetic data generator, while adding native support for handling missing values, outliers, and uninformative feature. Together, these properties make TabPFN-v2 a powerful probabilistic tabular regressor.

Our approach to time-series forecasting is complementary to standard TSFMs: rather than pretraining a new sequence model, we use TabPFN-v2 and reformulate forecasting as tabular regression via temporal featurization. Unlike most TSFMs—which are designed around autoregressive or `seq2seq` architectures—TabPFN-TS predicts the full horizon in a single non-autoregressive forward pass, offering a strong alternative to sequence-based foundation models.

## 3 Methodology

In this section, we present TabPFN-TS, a novel approach to using TabPFN-v2 for multi-step, univariate time-series forecasting. We recast time-series forecasting as a tabular regression problem, where a time series sequence is treated as a table, as shown in Figure 1.

### 3.1 From Time Series to Tabular Data

Given a time series $\mathbf{y}_{1:C+H} = [y_1, \ldots, y_{C+H}]$, the first $C$ observations define the historical context, and the remaining $H$ correspond to the forecast horizon. As illustrated in Figure 1, we construct feature matrices $\mathbf{X}_{\text{train}} \in \mathbb{R}^{C \times D}$ and $\mathbf{X}_{\text{test}} \in \mathbb{R}^{H \times D}$, where $D$ denotes the number of features. The corresponding historical targets form $\mathbf{y}_{\text{train}} \in \mathbb{R}^C$. This representation, consisting of $(\mathbf{X}_{\text{train}}, \mathbf{y}_{\text{train}})$ and $\mathbf{X}_{\text{test}}$, allows the use of any standard supervised tabular models, including TabPFN-v2.

---

[1]ChronosX and COSMIC are not publicly available at the time of writing and are therefore not included in our comparison.

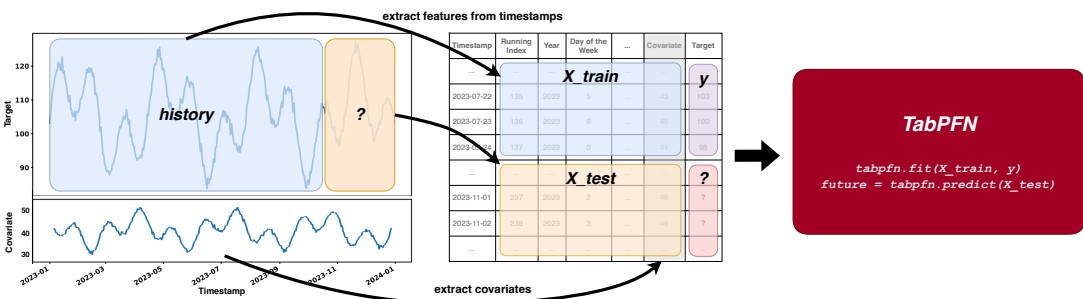

Figure 1: Overview of TabPFN-TS. From a given time series, we construct feature matrices $\mathbf{X}_{\text{train}}$ and $\mathbf{X}_{\text{test}}$, optionally appending exogenous covariates when available. The two matrices represent the historical and future windows, respectively. The observed historical targets form $\mathbf{y}_{\text{train}}$. TabPFN-v2 then conditions on the context $(\mathbf{X}_{\text{train}}, \mathbf{y}_{\text{train}})$ and predicts the corresponding targets at $\mathbf{X}_{\text{test}}$.

**Running Index.** To introduce a temporal reference within the timeline, we include the index of each time step as a feature (e.g., 0 for the first time step in the time series, 4 for the fifth):

$$\Phi_{\text{index}}(t) = t \in \mathbb{R}^1.$$

This provides a straightforward and effective way to track the progression of time across the observations.

**Calendar Features.** From each timestamp, we encode eight cyclic calendar components: second of minute, minute of hour, hour of day, day of week, day of month, day of year, week of year, and month of year. We additionally include the calendar year as a separate (non-cyclic) feature. Let $\{P_i\}_{i=1}^8$ be the periods associated with these cyclic components. The corresponding features for each timestamp $t$ are encoded as

$$\Phi_{\text{cal}}(t) = \left(\cos\left(\frac{2\pi t}{P_1}\right), \sin\left(\frac{2\pi t}{P_1}\right), \ldots, \cos\left(\frac{2\pi t}{P_8}\right), \sin\left(\frac{2\pi t}{P_8}\right), \text{year}(t)\right) \in \mathbb{R}^{17}$$

For full implementation details, see Appendix A.1.

**Automatic Seasonal Features.** Beyond the standard calendar periodicities, time series often have domain-specific cycles that calendar-based encodings fail to capture, e.g., depending on non-Gregorian calendars (Chinese birthdays) or the moon cycle (tides). To address this, we apply an automatic extraction process to identify the top-$k$ periodicities and encode them as features, thereby enriching the seasonality inputs to the model.

We first detrend each series via a simple linear least-squares regression. To reduce spectral leakage and improve frequency resolution, we apply a Hann window (Harris, 1978) and zero-pad the windowed signal by a factor of two (Oppenheim & Schafer, 1989). We then compute the real-valued discrete Fourier transform and remove the zero-frequency component of the spectrum, which corresponds to the mean of the time series and does not represent a seasonal oscillation. Finally, we select the $k$ largest spectral peaks by magnitude. Algorithm 1 provides high-level pseudo-code for this extraction process.

Given the detected frequencies $f_1, \ldots, f_k$, we then build the following features:

$$\Phi_{\text{auto}}(t) = (\cos(2\pi f_1 t), \ \sin(2\pi f_1 t), \ \ldots, \ \cos(2\pi f_k t), \ \sin(2\pi f_k t)) \in \mathbb{R}^{2k}.$$

**Time-varying Covariates (when available).** TabPFN-TS also supports time-varying covariates (although only those that are known for future points in time, such as holidays, not unknown ones like weather). We denote them by $\mathbf{z}_{1:C+H} = [\mathbf{z}_1, \ldots, \mathbf{z}_{C+H}]$, where each $\mathbf{z}_t \in \mathbb{R}^{D_z}$ represents $D_z$ external variables observed (or known) at time $t$.

For each timestamp, the full feature vector becomes:

$$\mathbf{x_t} = \Phi_{\text{index}}(t) \oplus \Phi_{\text{cal}}(t) \oplus \Phi_{\text{auto}}(t) \oplus \mathbf{z}_t \in \mathbb{R}^{(18+2k+D_z)}.$$

---

**Algorithm 1** Automatically extract top-$k$ Seasonalities, see Appendix A.2 for a detailed algorithm

---

**Require:** univariate series series, the number of periods to obtain $k$, the smoothing window size $L$

**Preprocess** series
Detrend linearly: $\text{series}[t] = \text{series}[t] - (\alpha t + \beta)$, where $\alpha$ and $\beta$ are found using least squares
Apply Hann window: $\text{series} = \text{conv}(\text{series}, w_{\text{Hann}}(L))$
Double length by symm. zero-padding: $\text{series} = [0, \ldots, 0, \text{series}[0], \ldots, \text{series}[N], 0, \ldots, 0]$

**Fourier Transform**
Compute FFT magnitudes mags and frequencies freqs based on the preprocessed series
Set $\text{mags}[0] = 0$ (remove the zero-frequency component)

**Select Peaks**
Find all peak indices peaks in mags (all (groups of) points larger than their neighbors)

**Convert & Clean**
Invert frequencies to periods and round $\text{periods} = \lfloor 1/\text{freqs} \rceil$
Remove duplicate and zero periods from the peak indices peaks
Keep only top $k$ peaks: $\text{peaks} = [i \text{ for } i \text{ in peaks if } i \text{ in topk}(\text{mags}[\text{peaks}])]$

**return** periods[peaks]

---

**Combining all Features.** Stacking these feature vectors across all timestamps yields the full feature matrix that describes the time series

$$\mathbf{X} = [\,\mathbf{x_1}; \ldots; \mathbf{x_{C+H}}\,] \in \mathbb{R}^{(C+H) \times D},$$

where $D = (18 + 2k + D_z)$ is the total number of features. Note that we do not rely on lagged or auto-regressive features (e.g., moving averages and lag terms), since these require past predictions and conflict with non-auto-regressive, multi-step forecasting making the inference much slower.

As described in Section 3.1, we split $\mathbf{X}$ into $\mathbf{X}_{\text{train}} = X_{1:C,:}$ and $\mathbf{X}_{\text{test}} = X_{C+1:C+H}$, corresponding to the historical and forecast windows.

### 3.2 Point and Probabilistic Forecasting with TabPFN-v2

We treat $(\mathbf{X}_{\text{train}}, \mathbf{y}_{\text{train}})$ as a classical regression dataset and feed it into TabPFN-v2. For each test input $x$ in $\mathbf{X}_{\text{test}}$, TabPFN-v2 outputs an approximate posterior predictive distribution $p(y \mid \mathbf{X}_{\text{train}}, \mathbf{y}_{\text{train}}, x)$.

In TabPFN-v2, this distribution is represented on a fixed numerical grid over the target space by assigning probabilities to bins (a discretized density). From this representation, we compute point predictions and probability summaries in the following way: the mean for squared-error evaluations, the median for absolute-error evaluations, and arbitrary quantiles (e.g. 5%, 50%, 95%) for probabilistic metrics and uncertainty bands.

## 4 Experiments

We present quantitative comparisons of TabPFN-TS against state-of-the-art forecasting models on both univariate (Section 4.1) and covariate-aware forecasting (Section 4.2) tasks. For all evaluations, we use a fixed configuration of TabPFN-v2 described in Appendix A.3.

### 4.1 Univariate Forecasting

We evaluate TabPFN-TS on GIFT-Eval (Aksu et al., 2024), a comprehensive benchmark developed to evaluate general time-series forecasting models.

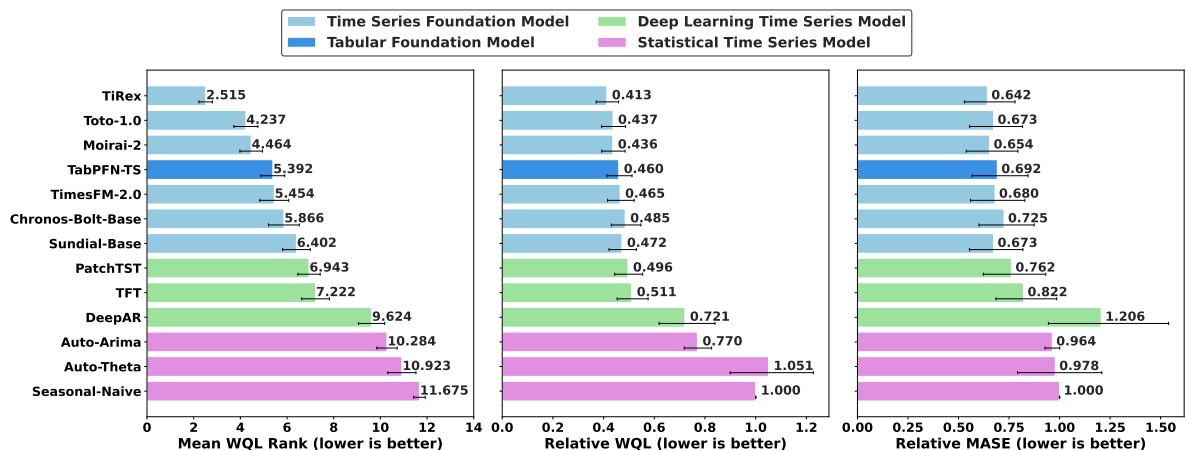

Figure 2: Univariate forecasting performance of TabPFN-TS and baseline models on GIFT-Eval benchmark. Although TabPFN-TS is not the top-ranked model, it achieves competitive performance on par with other time series foundation models (e.g. TimesFM2.0, Chronos-Bolt) while being a much smaller model (11M) and pretrained solely on tabular data. Scores are normalized by Seasonal Naive and aggregated across datasets; error bars show 95% confidence intervals.

**Datasets.** GIFT-Eval comprises 23 datasets with diverse characteristics, encompassing over $144,000$ time series and 177 million data points across seven application domains and ten different sampling frequencies. It covers both univariate and multivariate forecasting settings, as well as a wide range of prediction horizons, from short- to long-term forecasts. Considering all valid combinations of datasets, sampling frequencies, and prediction horizons, GIFT-Eval contains a total of 97 distinct benchmarking tasks. An overview of the datasets and their corresponding statistics is provided in Appendix A.4.

**Baselines.** We benchmark TabPFN-TS against a comprehensive set of baselines spanning statistical methods, deep learning models, and time series foundation models[2]. For classical statistical methods, we include Seasonal Naive, AutoETS, AutoARIMA, and AutoTheta (Garza et al., 2022). Among deep learning approaches, we evaluate DeepAR (Salinas et al., 2020) and Temporal Fusion Transformer (TFT) (Lim et al., 2021). For foundation models baselines, we include TiRex (Auer et al., 2025b), Toto-1.0 (Cohen et al., 2024), TimesFM-2.0 (Das et al., 2024), Chronos-Bolt (Ansari et al., 2024), and Moirai-2 (Woo et al., 2024).

**Evaluation Metrics.** Following prior work (Ansari et al., 2024; Shchur et al., 2023; Aksu et al., 2024), we assess point forecast accuracy using the Mean Absolute Scaled Error (MASE) and probabilistic forecast accuracy using the Weighted Quantile Loss (WQL). MASE normalizes the absolute forecast error by the historical seasonal error of each series, yielding a scale-invariant measure comparable across datasets. WQL evaluates the discrepancy between the predictive distribution and the observed value across quantile levels, providing a proxy assessment of probabilistic calibration. Consistent with GIFT-Eval, we compute WQL at uniformly spaced quantiles $\{0.1, 0.2, \ldots, 0.9\}$. Following Ansari et al. (2024), we aggregate relative scores across datasets using the geometric mean and additionally report the mean rank of WQL.

**Results.** Figure 2 summarizes the forecasting performance of TabPFN-TS on the GIFT-Eval benchmark. In both probabilistic forecasting (WQL) and point forecasting (MASE), TabPFN-TS ranks closely behind the leading foundation models TiRex, Toto-1.0, and Moirai-2, and matches substantially larger models such as TimesFM-2.0 and Chronos-Bolt, despite being over two orders of magnitude smaller and trained without any time-series–specific pretraining. Qualitative visualizations are provided in Appendix A.5 for reference.

These results demonstrate that TabPFN-TS produces accurate forecasts even though its backbone, TabPFN-v2, is pretrained solely on synthetic tabular data and relies on a simple temporal featurization. We addition-

---

[2]All baselines included were publicly available before September 2025, our literature cutoff date.

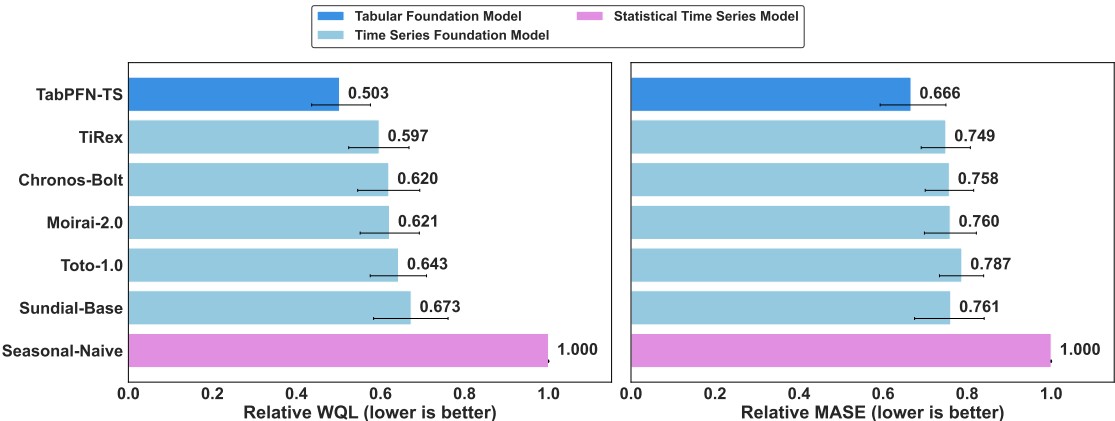

Figure 3: Covariate-informed forecasting performance of TabPFN-TS and baseline models on 28 `fev-bench` tasks with known dynamic covariates. As the only model that can directly incorporate covariate inputs, TabPFN-TS achieves the strongest results, outperforming all other models, including multivariate approaches such as Toto-1.0. Scores are normalized by Seasonal Naive and aggregated across datasets; error bars show 95% confidence intervals.

ally find that TabPFN-TS's predictive intervals are comparably calibrated to most time-series foundation model baselines (see Appendix A.10). This finding suggests that (i) the knowledge learned from generic tabular data can transfer effectively to temporal prediction, and (ii) further improvements are likely if tabular foundation models are exposed to real-world time-series data during pretraining. Together, these results position TabPFN-TS as a compact yet competitive alternative to large time-series foundation models.

## 4.2 Covariate-Informed Forecasting

For this task, we evaluate TabPFN-TS on `fev-bench` (Shchur et al., 2025), a general time-series forecasting benchmark that includes forecasting tasks where covariates are available.

**Datasets.** `fev-bench` comprises 100 forecasting tasks across multiple domains, of which 42 include covariate information. We focus on the 30 tasks where covariates are available in both the historical and future horizons. This configuration is required by TabPFN-TS, which models the dependency between the target series and known covariates and relies on having those covariate values available at prediction time. When future covariates are missing, the model cannot properly condition its forecasts on relevant contextual signals. Two of these tasks are excluded because TabPFN-TS exceeded the 6-hour evaluation limit, leaving 28 tasks for the final evaluation. Appendix A.6 provides an overview of the selected tasks.

**Baselines.** We follow the standardized evaluation protocol of `fev-bench`, which reports results only on pretrained time series foundation models. Deep learning baselines are not included here as `fev-bench` does not include dataset-specific supervised deep learning baselines. Therefore, like Section 4.1, TabPFN-TS compares against the following pretrained models: TiRex (Auer et al., 2025b), Chronos-Bolt (Base) (Ansari et al., 2024), Toto-1.0 (Cohen et al., 2024), Moirai-2.0 (Woo et al., 2024), and Sundial (Liu et al., 2025).

**Evaluation Metrics.** Consistently with Section 4.1, we assess point forecasting accuracy using the Mean Absolute Scaled Error (MASE) and probabilistic forecast accuracy using the Weighted Quantile Loss (WQL).

**Results.** Across the 28 `fev-bench` tasks with known dynamic covariates, TabPFN-TS achieves the strongest overall performance. While strong foundation models, such as TiRex and Chronos-Bolt, remain highly competitive in the univariate setting (see Section 4.1 and Shchur et al. (2025)), TabPFN-TS outperforms them when covariates are available. Notably, Toto-1.0, which is designed for the multivariate setting,

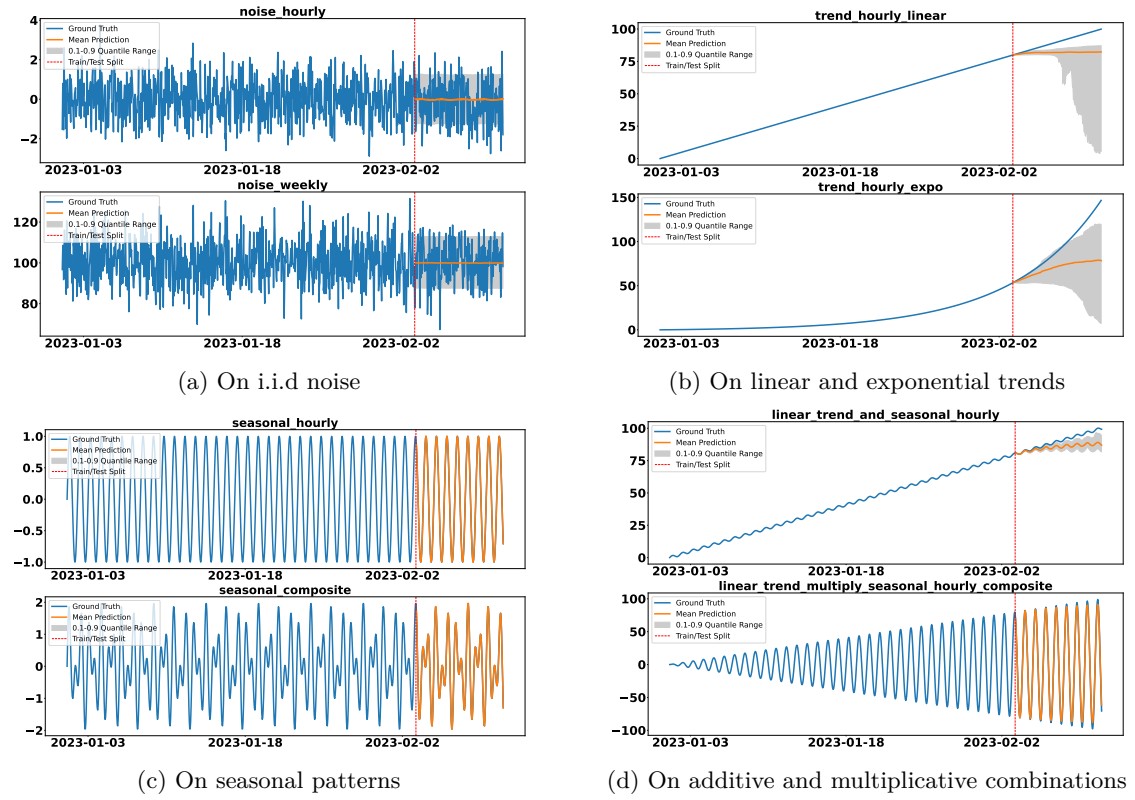

(a) On i.i.d noise

(b) On linear and exponential trends

(c) On seasonal patterns

(d) On additive and multiplicative combinations

Figure 4: Qualitative performance of TabPFN-v2 across noise, trend, and seasonal structures. The model handles noise and periodicity well but cannot extrapolate linear or exponential trend beyond the observed target range.

does not fully recover the benefit obtained when covariates are treated as first-class predictive features like in TabPFN-TS.

These results provide strong evidence for the forecasting-as-tabular-regression formulation (Section 3): it allows TabPFN-TS to integrate covariates directly and without architectural changes or fine-tuning. Together, this positions TabPFN-TS as a practical and competitive alternative to current time series foundation models, especially in settings where contextual or external drivers are known and informative.

## 5 Ablations

We conduct ablation studies to understand how TabPFN-TS performs forecasting and to assess the contribution of its components. Section 5.1 provides qualitative examples illustrating the model's forecasting behavior. Section 5.2 examines how the model interprets temporal structure within the tabular regression framework to perform forecasting. Section 5.3 studies the integration of covariate information under controlled synthetic setups. Section 5.4 evaluates alternative tabular regressors to assess the role of the pretrained backbone. Section 5.5 analyzes the impact of the temporal featurization design on overall performance.

### 5.1 Qualitative Analysis

We complement the quantitative benchmarks (Section 4) with qualitative evaluations on controlled synthetic series, following the setups proposed by Ansari et al. (2024). In all cases, the first 800 points are provided as context and the model forecasts the next 200 steps. These experiments illustrate the characteristic strengths and limitations of TabPFN-TS in isolation from dataset-specific confounders.

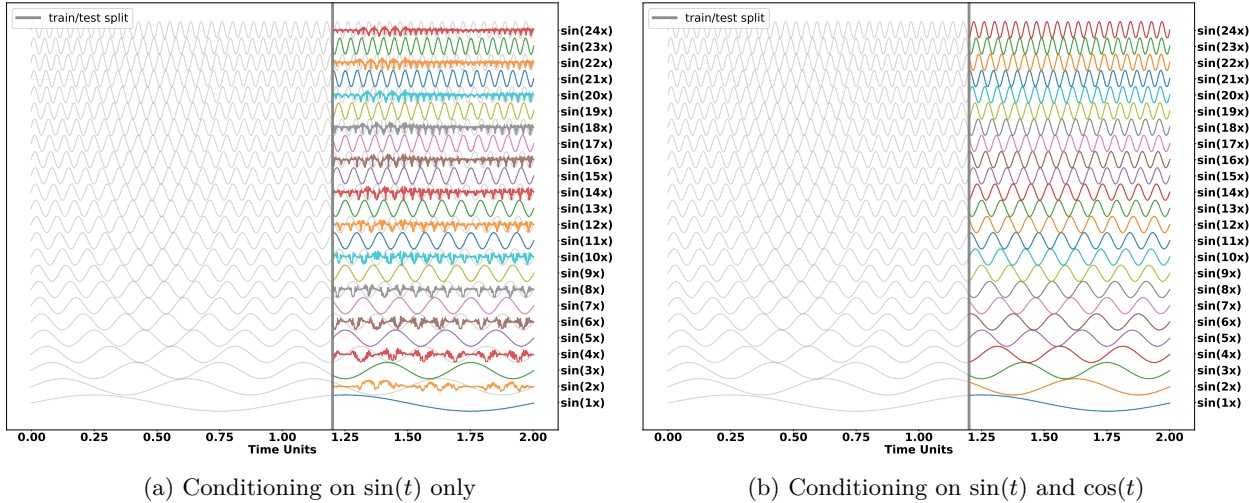

(a) Conditioning on $\sin(t)$ only

(b) Conditioning on $\sin(t)$ and $\cos(t)$

Figure 5: Reconstruction of sinusoidal signals under different temporal embeddings. TabPFN-v2 is evaluated on $\sin(nt)$ for $n = 1, \ldots, 24$. (a) With $\sin(t)$ as the sole temporal feature, the non-injective embedding causes different time points to collapse to the same feature representation, and the model therefore reconstructs odd but not even harmonics. (b) With $\{\sin(t), \cos(t)\}$, the injective embedding allows the model to interpolate reliably in this phase space and reconstruct all harmonics. These results suggest that periodic generalization arises from interpolation in an expressive temporal embedding.

**I.I.D. Noise.** Figure 4a shows the model's predictions when the input is i.i.d. noise sampled from $\mathcal{N}(0, 1)$ and $\mathcal{N}(100, 10)$ at hourly and weekly resolution. TabPFN-TS does not overfit: it predicts the sample mean and produces predictive intervals that closely align with the corresponding Gaussian quantiles. This demonstrates stable behavior and reasonable uncertainty when no temporal structure is present.

**Seasonality.** TabPFN-TS models seasonal structure exceptionally well (Figures 4c–d). When periodic patterns are represented in the features, the model not only reconstructs the signal accurately but also produces very tight predictive intervals that closely follow the ground truth. This strong behaviour is further discussed in Section 5.2, where we show that the sinusoidal embedding provides an expressive phase representation that enables precise interpolation of periodic components.

**Trend.** Figure 4b illustrates a clear limitation: TabPFN-TS does not extrapolate trends when future values fall outside the range of the conditioning set (i.e. beyond the observed target range). This behavior appears both for linear and exponential trends. Since the model operates via interpolation in feature space (discussed later in Section 5.2), it cannot extend predictions beyond the observed target domain—an important limitation for time series with sustained growth or decay.

**Takeaway.** These qualitative results reveal a consistent pattern: TabPFN-TS behaves reliably when the future targets lie within the range of the conditioning set—capturing noise and seasonal patterns with accurate means and tight uncertainty intervals that align well with the underlying data distribution—but fails to extrapolate when forecasting requires moving beyond the observed target domain, as in linear or exponential trends. This contrast highlights that the model's strengths and limitations depend heavily on how temporal structure is represented, motivating the deeper analysis in Section 5.2 and Section 6.1.

## 5.2 How TabPFN-TS Interprets Temporal Structure

To understand why TabPFN-v2 handles seasonality effectively (see Section 5.1) despite being pretrained only on tabular data, we analyze how its backbone (TabPFN-v2) responds to minimal temporal featurization when forecasting periodic signals. This reveals the precise mechanism by which the model leverages temporal cues.

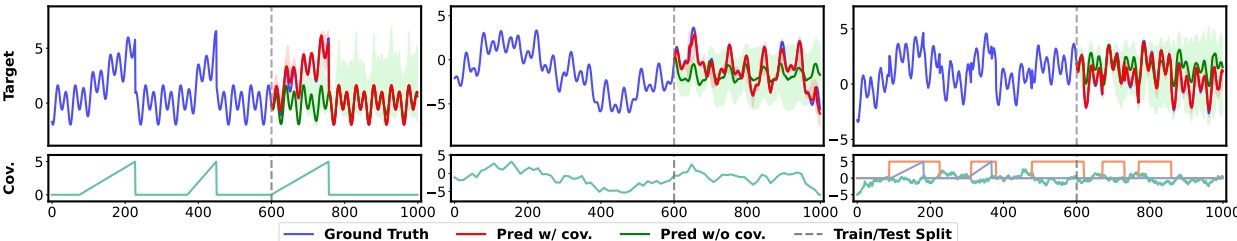

Figure 6: Forecasts of TabPFN-TS on synthetic time series augmented by time-varying covariates. Median predictions are shown in red (conditioned on covariates) and green (unconditioned), with shaded bands indicating the $10^{th}$ - $90^{th}$ percentiles.

**Setup.** We generate target series of the form $y_t = \sin(nt)$ for $n = 1, \ldots, 24$. The model is evaluated under two featurization settings: (i) using $\sin(t)$ alone, and (ii) the pair $(\sin(t), \cos(t))$. This enables us to test whether the model can recover higher-order harmonics from a low-frequency base. We then extend this to composite signals formed by summing multiple sinusoids with random frequencies, amplitudes, and phases.

**Findings.** With only $\sin(t)$, the model reconstructs odd harmonics but fails entirely on even ones (Figure 5a). However, with the pair $(\sin(t), \cos(t))$, the model accurately recovers the higher-order harmonics.

We hypothesize that this behavior stems from the **injectivity** of the temporal mapping. When only $\sin(t)$ is present, the mapping $t \mapsto \sin(t)$ is not injective; distinct time points (e.g., on the rising vs. falling edges of a cycle) share identical feature values, making local interpolation inherently ambiguous. The inclusion of $\cos(t)$ embeds time onto the unit circle, creating an injective mapping that uniquely identifies the phase (see Section A.7.1 for a detailed symmetry analysis). Under this representation, TabPFN-v2 accurately recovers harmonics across a wide frequency range (Figure 5b), and successfully reconstructs complex composite signals (Figure 16).

We also observe a gradual deterioration in accuracy as the frequency increases (Figure 14 and 15). This is consistent with the finite resolution of the temporal embedding and classical aliasing effect (Oppenheim, 1999): as frequency increases, the angular distance between successive samples in the coordinate system decreases, making adjacent phases increasingly difficult to be distinguished.

While we considered an alternative hypothesis where the model performs global functional approximation (e.g., via Chebyshev expansion), our preliminary experiments (see Appendix A.7.2) suggest that TabPFN-v2's behavior is more consistent with a retrieval mechanism.

**Takeaway.** These results suggest that when supplied with an injective sinusoidal embedding of time, TabPFN-TS behaves as a **meta-learned retrieval engine**. It interpolates effectively within a feature space where time has been mapped to a unique circular coordinate (i.e. phase-space), enabling phase-aware pattern matching against the observed history in the support set.

## 5.3 How TabPFN-TS Utilizes Covariates

In Section 4.2, we showed that TabPFN-v2 achieves the strongest performance on covariate-informed forecasting tasks. To better understand this effect, we conduct controlled synthetic experiments where the informativeness and interaction structure of each covariate are known.

**Synthetic Setup.** Similar to Arango et al. (2025) and Auer et al. (2025a), we generate synthetic time series where the relationship between the target and the covariates is fully controlled. Each series consists of a stationary stochastic process (Brockwell & Davis, 2016)—implemented here as a sum of sinusoids with random amplitudes, phases, and periods—augmented with one or more covariates that follow common temporal patterns such as linear trends, pulses, ramps, or stochastic drifts. Covariates influence the target through either additive or multiplicative coupling, resulting in ten interpretable synthetic classes (Figure 18).

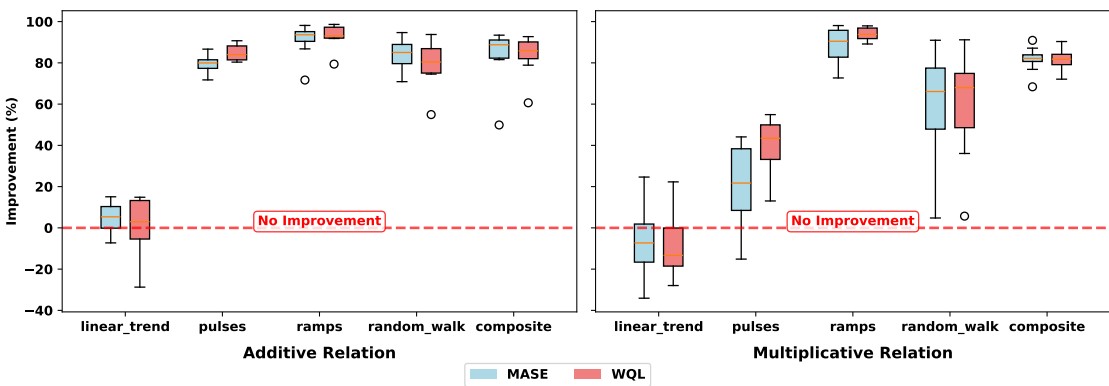

Figure 7: Improvement when incorporating covariates across synthetic classes. Boxplots show relative improvement (in percentage, %) in MASE (blue) and WQL (red). Most covariate types lead to substantial improvement, particularly for pulses, ramps, random walks, and composite signals. In contrast, for both additive and multiplicative linear trend series, covariates fail to improve forecasts, reflecting the model's limitation in extrapolating linear growth.

**Evaluation Protocol.** For each class, we evaluate TabPFN-TS under two conditions: (i) using temporal features only, and (ii) using both temporal features and covariates. This isolates the incremental contribution of covariate information.

**Findings.** Figure 7 shows that incorporating covariates substantially improves forecasting accuracy for most synthetic classes—pulses, ramps, random walks, and composite signals—reducing both MASE and WQL and producing sharper uncertainty estimates (Figure 18). These gains are consistent with our benchmark results and indicate that TabPFN-TS can reliably exploit informative auxiliary signals when their effects lie within the range covered by the conditioning set (detailed in Section 6.1).

The main exception is the *linear-trend* setting, where covariates provide little benefit because the future targets fall outside the distribution observed in the context; TabPFN-TS does not extrapolate linear growth (discussed later in Section 6.1). Overall, these controlled experiments show that TabPFN-TS effectively leverages covariates across a wide range of scenarios, with extrapolation remaining the primary limitation.

## 5.4 Choice of Tabular Backbone

To assess whether our forecasting formulation depends on the choice of tabular backbone, we evaluate two categories of alternative backbones under the same temporal featurization: (i) classical tabular regressors XGBoost (Chen & Guestrin, 2016), LightGBM (Ke et al., 2017), and CatBoost (Prokhorenkova et al., 2018); and (ii) TabDPT (Ma et al., 2025), one of the few open-source tabular foundation models that supports regression (Erickson et al., 2025). As most of these models only produce point predictions, we evaluate on point forecasting (MASE) across 61 of the 97 GIFT-Eval tasks (the remainder exceeded our compute budget).

Figure 8 shows that all the alternative backbones achieve meaningful forecasting performance, yet fall short of TabPFN-TS. We attribute this to the pretraining: TabPFN-v2 TabPFN-v2 is trained on millions of synthetic tasks spanning diverse functional relationships and noise regimes, while the gradient-boosted mod-

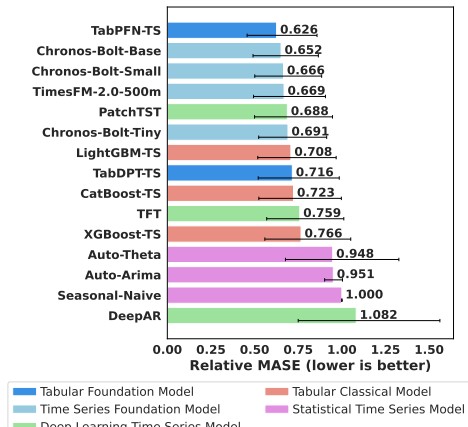

Figure 8: MASE comparison of different tabular backbones on 67 GIFT-Eval tasks. TabPFN-TS outperforms all alternatives by a clear margin, suggesting TabPFN-v2's synthetic pretraining contributes beyond featurization alone.

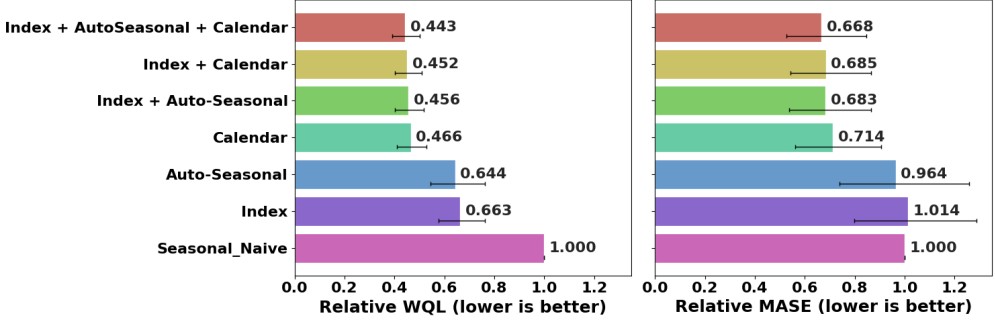

Figure 9: Ablation of temporal featurization components in TabPFN-TS. Relative WQL (left) and MASE (right) show that all components contribute meaningfully to performance. Using only the index or only the auto-seasonal features yields substantially worse results, while combining all feature types achieves the best accuracy, underscoring the value of the proposed featurization scheme.

els are fit from scratch per series and TabDPT is pretrained on real-world tabular data with narrower task diversity.

Taken together, this experiment suggests that: (i) the **temporal featurization generalizes across tabular foundation models**—different tabular foundation models can perform zero-shot forecasting under the same representation; and (ii) the **regression backbone matters**—models pretrained on broader and more diverse task distributions (such as TabPFN-v2) appear to transfer more effectively to the time-series forecasting.

### 5.5 Impact of Temporal Featurization

To assess the contribution of the each temporal feature type (Section 3.1), we evaluate TabPFN-TS under different combinations of these features on the 81 smallest tasks (out of 97) in GIFT-Eval.

The results in Figure 9 highlight the importance of **encoding time progression**. Without an explicit index, calendar features provide both periodic structure and a weak notion of temporal progression, since some calendar attributes present monotonically within the timeline. Automatic seasonal features, by contrast, encode oscillatory components but offer no information about global temporal progression. When the index is present, it provides the primary temporal axis, while both calendar and automatic seasonal features contribute the complementary periodic structure.

The best results are obtained when all feature types are combined. This indicates that calendar and automatic seasonal features capture different—and complementary—seasonal patterns. Together with the index, they provide a broad and expressive temporal representation.

Overall, TabPFN-TS benefits from temporal features that jointly encode **monotonic progression** and **multi-scale seasonality**. The framework is naturally extensible, allowing practitioners to incorporate domain-specific periodic features to further improve forecasting accuracy when such knowledge is available.

## 6 Limitation

### 6.1 Extrapolation Beyond the Conditioning Range

The analyses in Sections 5.1 and 5.3 reveal a persistent limitation of TabPFN-TS: it struggles in extrapolating trends, where the future target values lie outside the range observed in the conditioning set.

This behavior is a direct consequence of how TabPFN-v2 is pretrained. During pretraining, each synthetic task is generated by sampling a tabular dataset and randomly partitioning it into a context set $(X, y)$ and a query set $X'$. The model is then trained to predict the corresponding targets $y'$. Since both context and query are sampled from the same generating distribution, the query targets typically lie within the convex

hull of the observed targets. Consequently, TabPFN-v2 is primarily optimized for conditional interpolation and lacks the inductive bias required for unbounded trend extrapolation.

TabPFN-TS inherits this inductive bias: when future targets lie outside the range of observed target, it has little basis—under its interpolation-oriented objective—to project the trajectory further. Predictions therefore remain within or close to the span of the observed targets.

In summary, TabPFN-v2 is a strong conditional interpolator but lacks a mechanism for systematic trend extrapolation. This limitation is structural, stemming from the pretraining objective, rather than from the tabular regression formulation used for forecasting.

## 6.2 Inference Time

A second practical limitation of TabPFN-TS is its higher computational cost at inference. Although the model supports data-parallel evaluation across multiple GPUs, this only increases throughput but does not reduce the per-time-series cost. Figure 10 reports single-GPU-equivalent inference times, normalized by GPU count to allow direct comparison with Chronos-Bolt-Base and TimesFM-2.0. Under equal hardware assumptions, TabPFN-TS requires roughly $30\times$ more inference time.

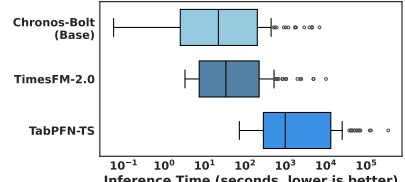

Figure 10: Per-task inference time on 97 GIFT-Eval tasks (log scale; lower is better). Chronos-Bolt-Base and TimesFM-2.0 run on one NVIDIA T4; TabPFN-TS runs across four T4s, and we report time normalized to one GPU.

This overhead arises from how TabPFN-v2 performs prediction. In tabular settings, one forward pass produces the prediction for an entire dataset. In time-series forecasting, however, a dataset contains many individual series—often hundreds or thousands—and each series becomes an independent query instance under our formulation. TabPFN-TS must therefore execute a separate forward pass for every series.

Moreover, TabPFN-TS retains the full-attention architecture of TabPFN-v2, which incurs $O(L^2)$ computational complexity in sequence length $L$. Recent time-series foundation models mitigate this cost by processing large batches of series and by using architectural techniques such as patching or lightweight decoders that reduce dependence on sequence length. Consequently, they achieve substantially faster inference.

In summary, the higher inference cost of TabPFN-TS arises structurally from the combination of (i) its one-forward-pass-per-series formulation and (ii) its quadratic attention mechanism. Improving inference efficiency—through patching, batching, and/or architectural refinement (discussed in Section A.11)—remains an important direction for future work.

## 7 Conclusion

**Summary of Contributions.** This work demonstrates that a tabular foundation model—pretrained solely on synthetic, non-temporal data—can be adapted into a competitive zero-shot forecaster through a simple reformulation of time-series prediction as tabular regression. Combining TabPFN-v2 with a lightweight temporal featurization yields TabPFN-TS, a compact (11M) model that requires no time-series–specific pretraining or fine-tuning, yet achieves strong performance on univariate forecasting and state-of-the-art accuracy on covariate-informed tasks.

**Key Empirical Insights.** Our ablations highlight three central findings. (1) Temporal featurization is essential: performance improves substantially when monotonic progression and complementary periodicities are jointly encoded. (2) Pretraining induces distinct inductive biases: TabPFN-TS excels at interpolation inside the observed target range, but—consistent with its pretraining objective—fails when forecasting requires systematic extrapolation. (3) Forecasting capability is model-agnostic: substituting the backbone with Tab-DPT still produces meaningful zero-shot forecasts, indicating that the overall formulation generalizes across tabular foundation models.

**Implications.** These results indicate that tabular foundation models constitute a surprisingly effective and data-efficient alternative to specialized time-series architectures. The tabular formulation naturally unifies univariate and covariate-aware forecasting, treats all inputs as general features, and enables probabilistic forecasting without architectural modifications to the backbone.

**Vision and Future Directions.** More broadly, this work suggests a promising direction toward unified foundation models for structured data that operate seamlessly over tabular, temporal, and hybrid inputs. Addressing the core limitation—the inability to extrapolate trends beyond the conditioning range—may require new pretraining objectives that blend interpolation, controlled extrapolation, and explicit handling of out-of-support targets. Such advances open a path toward compact, domain-agnostic predictors that bridge long-standing gaps between tabular learning and time-series forecasting.

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

# A  Appendix

## A.1  Implementation of Calendar Features

---

**Algorithm 2** Detailed Calendar Features Implementation

---

**Require:**
    Time-indexed table $\mathcal{D}$ with index level `timestamp`.

**Ensure:**
    $\mathcal{D}$ augmented with:

-  `year` column;
-  sin and cos embeddings for each of:
  $\big(\texttt{second\_of\_minute}, 60\big)$,   $\big(\texttt{minute\_of\_hour}, 60\big)$,   $\big(\texttt{hour\_of\_day}, 24\big)$,   $\big(\texttt{day\_of\_week}, 7\big)$,
  $\big(\texttt{day\_of\_month}, 30.5\big)$, $\big(\texttt{day\_of\_year}, 365\big)$, $\big(\texttt{week\_of\_year}, 52\big)$, $\big(\texttt{month\_of\_year}, 12\big)$.

1:  $\mathcal{D} \leftarrow \mathcal{D}.\text{copy}()$
2:  $\mathbf{T} \leftarrow \mathcal{D}.\text{index.get\_level\_values}(\texttt{"timestamp"})$

    **Extract year component**
3:  $\mathcal{D}[\texttt{"year"}] \leftarrow \mathbf{T}.\text{year}$

    **Extract calendar-based seasonality**
4:  $\mathcal{S} \leftarrow \{$
      $(\texttt{"second\_of\_minute"}, 60),$
      $(\texttt{"minute\_of\_hour"}, 60),$
      $(\texttt{"hour\_of\_day"}, 24),$
      $(\texttt{"day\_of\_week"}, 7),$
      $(\texttt{"day\_of\_month"}, 30.5),$
      $(\texttt{"day\_of\_year"}, 365),$
      $(\texttt{"week\_of\_year"}, 52),$
      $(\texttt{"month\_of\_year"}, 12)$
    $\}$                ▷ List of seasonal features with their natural periods

5:  **for all** $(name, P)$ in $\mathcal{S}$ **do**
6:     $\mathbf{f} \leftarrow \text{time\_feature}(name).\text{index}(\mathbf{T})$             ▷ integer cycle index
7:     $\tilde{P} \leftarrow P - 1$
8:     $\mathcal{D}[name \| \texttt{\_sin}] \leftarrow \sin\big(2\pi \, \mathbf{f} / \tilde{P}\big)$
9:     $\mathcal{D}[name \| \texttt{\_cos}] \leftarrow \cos\big(2\pi \, \mathbf{f} / \tilde{P}\big)$
10:  **end for**
11:  **return** $\mathcal{D}$

---

## A.2 Implementation of Automatic Seasonal Features

---

**Algorithm 3** Detailed Extract top-$k$ Seasonalities Algorithm

---

**Require:**
- Time series $\mathbf{x}_t = \{x_1, x_2, \ldots, x_N\}$
- Integer $k$ (max number of periods)
- Hann window length $L$

**Ensure:** Set $\mathcal{P}$ of up to $k$ dominant periods

    **Preprocessing:**

1: Detrend $\mathbf{x}_t$ via linear regression

$$\tilde{x}_t = x_t - (\alpha t + \beta) \quad , \text{ where } \alpha \text{ and } \beta \text{ are found using least squares}$$

2: Apply Hann window:

$$w'_t = 0.5 \left(1 - \cos\left(\frac{2\pi t'}{L}\right)\right) \quad \text{for} \quad t \in \{0, \ldots, L\}$$

$$\breve{x} = conv(\tilde{x}, w)$$

3: Symmetrically zero-pad to length $2N$:

$$\mathbf{y} = [0, \ldots, 0, \breve{x}_1, \ldots, \breve{x}_N, 0, \ldots, 0]$$

    **Spectral Analysis:**

4: Compute fast fourier transform:

$$Y_k = \sum_{t=1}^{2N} y_t e^{-i2\pi(k-1)t/(2N)} \quad \text{for } k = 1, \ldots, N$$

$$\text{Magnitudes: } A_k = |Y_k|$$

$$\text{Frequencies: } f_k = \frac{k-1}{2N} \quad \text{(normalized to Nyquist)}$$

5: Remove DC component:

$$A_1 \leftarrow 0$$

    **Peak Selection:**

6: Identify local maxima (peaks larger than immediate neighbors, taking midpoint of multi-point peaks in practice):

$$\mathcal{L} = \left\{i \in \{2, \ldots, N-1\} \,\middle|\, A_i > A_{i-1} \text{ and } A_i > A_{i+1}\right\}$$

    **Period Conversion:**

7: Convert frequencies to periods and round to integers:

$$p_i = \left\lfloor \frac{1}{f_i} \right\rceil, \text{ for } i \in L$$

8: Remove duplicates and 0. periods, yielding a new set of indexes $\mathcal{I}$
9: Build top $k$ index set $\mathcal{T} = \{i \in \mathcal{I} | A_i \in \text{topk}_k(\{A_i | i \in \mathcal{I}\})\}$
10: **return** $\{p_i | i \in \mathcal{T}\}$

---

### A.3   TabPFN-TS Configuration

We describe the configuration used for all TabPFN-TS experiments. Unless otherwise noted, all settings are fixed across benchmarks.

**Data Preprocessing.**   For time series with missing observations, we remove the affected time points from the conditioning set. For efficiency, we restrict the conditioning window to the most recent 4096 observations (after removing missing points), which we found to provide a good balance between accuracy and computational cost (see Appendix A.9). No additional normalization is applied to the target values, as TabPFN-v2 performs internal z-normalization during inference.

**TabPFN-v2 Model Configuration.**   TabPFN-v2 offers several publicly available regression checkpoints. For all experiments, we use the `tabpfn-v2-regression-2noar4o2.ckpt`[3], which showed stable behavior across both point and probabilistic forecasting in preliminary evaluations. We do not perform hyperparameter tuning or checkpoint selection beyond this fixed choice, as our focus is on evaluating the utility of the forecasting formulation rather than optimizing the backbone.

**Temporal Featurization.**   We apply the full featurization pipeline described in Section 3.1. For Automatic Seasonal Features, we retain the top $k = 5$ periodicities identified by the spectral analysis described in Section 3.1. This value provides a good balance between expressiveness and computational efficiency: increasing $k$ adds additional feature dimensions, but we found in preliminary checks that the model's forecasting accuracy is relatively stable across reasonable values of $k$. A more extensive study of this parameter is left for future work.

---

[3]https://huggingface.co/Prior-Labs/TabPFN-v2-reg/blob/9ff6a7be140b2663cdd21387a5decc3a8018ad6b/tabpfn-v2-regression-2noar4o2.ckpt

## A.4 GIFT-Eval Benchmark Datasets and Corresponding Statistics

Each benchmarking task in GIFT-Eval corresponds to a unique combination of dataset, prediction horizon (short-, medium-, or long-term), and sampling frequency (where applicable). For a given dataset, a benchmarking task is defined only if sufficient historical data is available to support the specified window size and forecast length, as shown in the short-, medium-, and long-term columns of Table 1. In total, GIFT-Eval comprises 97 such tasks that span diverse domains, temporal resolutions, and forecasting lengths.

These 97 tasks are used in the main experimental evaluation. For the ablation studies, we exclude datasets marked with an asterisk (*) due to their relatively large size and higher resource requirements.

Table 1: Statistics of datasets from the GIFT-Eval benchmark (reproduced from Aksu et al. (2024) under a CC BY 4.0 license). Datasets marked with an asterisk (*) are excluded from ablation studies due to the large size.

| Dataset | Domain | Frequency | # Series | Series Length Avg | Series Length Min | Series Length Max | # Obs | # Target Variates | Short-term Pred Length | Short-term Windows | Med-term Pred Length | Med-term Windows | Long-term Pred Length | Long-term Windows |
|---|---|---|---|---|---|---|---|---|---|---|---|---|---|---|
| Jena Weather | Nature | 10T | 1 | 52,704 | 52,704 | 52,704 | 52,704 | 21 | 48 | 20 | 480 | 11 | 720 | 8 |
| Jena Weather | Nature | H | 1 | 8,784 | 8,784 | 8,784 | 8,784 | 21 | 48 | 19 | 480 | 2 | 720 | 2 |
| Jena Weather | Nature | D | 1 | 366 | 366 | 366 | 366 | 21 | 30 | 2 | | | | |
| BizITObs - Application | Web/CloudOps | 10S | 1 | 8,834 | 8,834 | 8,834 | 8,834 | 2 | 60 | 15 | 600 | 2 | 900 | 1 |
| BizITObs - Service | Web/CloudOps | 10S | 21 | 8,835 | 8,835 | 8,835 | 185,535 | 2 | 60 | 15 | 600 | 2 | 900 | 1 |
| BizITObs - L2C | Web/CloudOps | 5T | 1 | 31,968 | 31,968 | 31,968 | 31,968 | 7 | 48 | 20 | 480 | 7 | 720 | 5 |
| BizITObs - L2C | Web/CloudOps | H | 1 | 2,664 | 2,664 | 2,664 | 2,664 | 7 | 48 | 6 | 480 | 1 | 720 | 1 |
| Bitbrains - Fast Storage | Web/CloudOps | 5T* | 1,250 | 8,640 | 8,640 | 8,640 | 10,800,000 | 2 | 48 | 18 | 480 | 2 | 720 | 2 |
| Bitbrains - Fast Storage | Web/CloudOps | H | 1,250 | 721 | 721 | 721 | 901,250 | 2 | 48 | 2 | | | | |
| Bitbrains - rnd* | Web/CloudOps | 5T | 500 | 8,640 | 8,640 | 8,640 | 4,320,000 | 2 | 48 | 18 | 480 | 2 | 720 | 2 |
| Bitbrains - rnd | Web/CloudOps | H | 500 | 720 | 720 | 720 | 360,000 | 2 | 48 | 2 | | | | |
| Restaurant | Sales | D | 807 | 358 | 67 | 478 | 289,303 | 1 | 30 | 1 | | | | |
| ETT1 | Energy | 15T | 1 | 69,680 | 69,680 | 69,680 | 69,680 | 7 | 48 | 20 | 480 | 15 | 720 | 10 |
| ETT1 | Energy | H | 1 | 17,420 | 17,420 | 17,420 | 17,420 | 7 | 48 | 20 | 480 | 4 | 720 | 3 |
| ETT1 | Energy | D | 1 | 725 | 725 | 725 | 725 | 7 | 30 | 3 | | | | |
| ETT1 | Energy | W-THU | 1 | 103 | 103 | 103 | 103 | 7 | 8 | 2 | | | | |
| ETT2 | Energy | 15T | 1 | 69,680 | 69,680 | 69,680 | 69,680 | 7 | 48 | 20 | 480 | 15 | 720 | 10 |
| ETT2 | Energy | H | 1 | 17,420 | 17,420 | 17,420 | 17,420 | 7 | 48 | 20 | 480 | 4 | 720 | 3 |
| ETT2 | Energy | D | 1 | 725 | 725 | 725 | 725 | 7 | 30 | 3 | | | | |
| ETT2 | Energy | W-THU | 1 | 103 | 103 | 103 | 103 | 7 | 8 | 2 | | | | |
| Loop Seattle* | Transport | 5T | 323 | 105,120 | 105,120 | 105,120 | 33,953,760 | 1 | 48 | 20 | 480 | 20 | 720 | 15 |
| Loop Seattle* | Transport | H | 323 | 8,760 | 8,760 | 8,760 | 2,829,480 | 1 | 48 | 19 | 480 | 2 | 720 | 2 |
| Loop Seattle | Transport | D | 323 | 365 | 365 | 365 | 117,895 | 1 | 30 | 2 | | | | |
| SZ-Taxi | Transport | 15T | 156 | 2,976 | 2,976 | 2,976 | 464,256 | 1 | 48 | 7 | 480 | 1 | 720 | 1 |
| SZ-Taxi | Transport | H | 156 | 744 | 744 | 744 | 116,064 | 1 | 48 | 2 | | | | |
| M_DENSE | Transport | H | 30 | 17,520 | 17,520 | 17,520 | 525,600 | 1 | 48 | 20 | 480 | 4 | 720 | 3 |
| M_DENSE | Transport | D | 30 | 730 | 730 | 730 | 21,900 | 1 | 30 | 3 | | | | |
| Solar | Energy | 10T | 137 | 52,560 | 52,560 | 52,560 | 7,200,720 | 1 | 48 | 20 | 480 | 11 | 720 | 8 |
| Solar | Energy | H | 137 | 8,760 | 8,760 | 8,760 | 1,200,120 | 1 | 48 | 19 | 480 | 2 | 720 | 2 |
| Solar | Energy | D | 137 | 365 | 365 | 365 | 50,005 | 1 | 30 | 2 | | | | |
| Solar | Energy | W-FRI | 137 | 52 | 52 | 52 | 7,124 | 1 | 8 | 1 | | | | |
| Hierarchical Sales | Sales | D | 118 | 1,825 | 1,825 | 1,825 | 215,350 | 1 | 30 | 7 | | | | |
| Hierarchical Sales | Sales | W-WED | 118 | 260 | 260 | 260 | 30,680 | 1 | 8 | 4 | | | | |
| M4 Yearly | Econ/Fin | A-DEC | 22,974 | 37 | 19 | 284 | 845,109 | 1 | 6 | 1 | | | | |
| M4 Quarterly | Econ/Fin | Q-DEC | 24,000 | 100 | 24 | 874 | 2,406,108 | 1 | 8 | 1 | | | | |
| M4 Monthly | Econ/Fin | M | 48,000 | 234 | 60 | 2,812 | 11,246,411 | 1 | 18 | 1 | | | | |
| M4 Weekly | Econ/Fin | W-SUN | 359 | 1,035 | 93 | 2,610 | 371,579 | 1 | 13 | 1 | | | | |
| M4 Daily | Econ/Fin | D | 4,227 | 2,371 | 107 | 9,933 | 10,023,836 | 1 | 14 | 1 | | | | |
| M4 Hourly | Econ/Fin | H | 414 | 902 | 748 | 1,008 | 373,372 | 1 | 48 | 2 | | | | |
| Hospital | Healthcare | M | 767 | 84 | 84 | 84 | 64,428 | 1 | 12 | 1 | | | | |
| COVID Deaths | Healthcare | D | 266 | 212 | 212 | 212 | 56,392 | 1 | 30 | 1 | | | | |
| US Births | Healthcare | D | 1 | 7,305 | 7,305 | 7,305 | 7,305 | 1 | 30 | 20 | | | | |
| US Births | Healthcare | W-TUE | 1 | 1,043 | 1,043 | 1,043 | 1,043 | 1 | 8 | 14 | | | | |
| US Births | Healthcare | M | 1 | 240 | 240 | 240 | 240 | 1 | 12 | 2 | | | | |
| Saugeen | Nature | D | 1 | 23,741 | 23,741 | 23,741 | 23,741 | 1 | 30 | 20 | | | | |
| Saugeen | Nature | W-THU | 1 | 3,391 | 3,391 | 3,391 | 3,391 | 1 | 8 | 20 | | | | |
| Saugeen | Nature | M | 1 | 780 | 780 | 780 | 780 | 1 | 12 | 7 | | | | |
| Temperature Rain* | Nature | D | 32,072 | 725 | 725 | 725 | 780 | 1 | 30 | 3 | | | | |
| KDD Cup 2018 | Nature | H | 270 | 10,898 | 9,504 | 10,920 | 2,942,364 | 1 | 48 | 20 | 480 | 2 | 720 | 2 |
| KDD Cup 2018 | Nature | D | 270 | 455 | 396 | 455 | 122,791 | 1 | 30 | 2 | | | | |
| Car Parts | Sales | M | 2,674 | 51 | 51 | 51 | 136,374 | 1 | 12 | 1 | | | | |
| Electricity* | Energy | 15T | 370 | 140,256 | 140,256 | 140,256 | 51,894,720 | 1 | 48 | 20 | 480 | 20 | 720 | 20 |
| Electricity | Energy | H | 370 | 35,064 | 35,064 | 35,064 | 12,973,680 | 1 | 48 | 20 | 480 | 8 | 720 | 5 |
| Electricity | Energy | D | 370 | 1,461 | 1,461 | 1,461 | 540,570 | 1 | 30 | 5 | | | | |
| Electricity | Energy | W-FRI | 370 | 208 | 208 | 208 | 76,960 | 1 | 8 | 3 | | | | |

## A.5  Additional Results on GIFT-Eval

Figures 11–13 present example predictions from TabPFN-TS on randomly selected samples from short-, medium-, and long-term forecasting tasks, respectively.

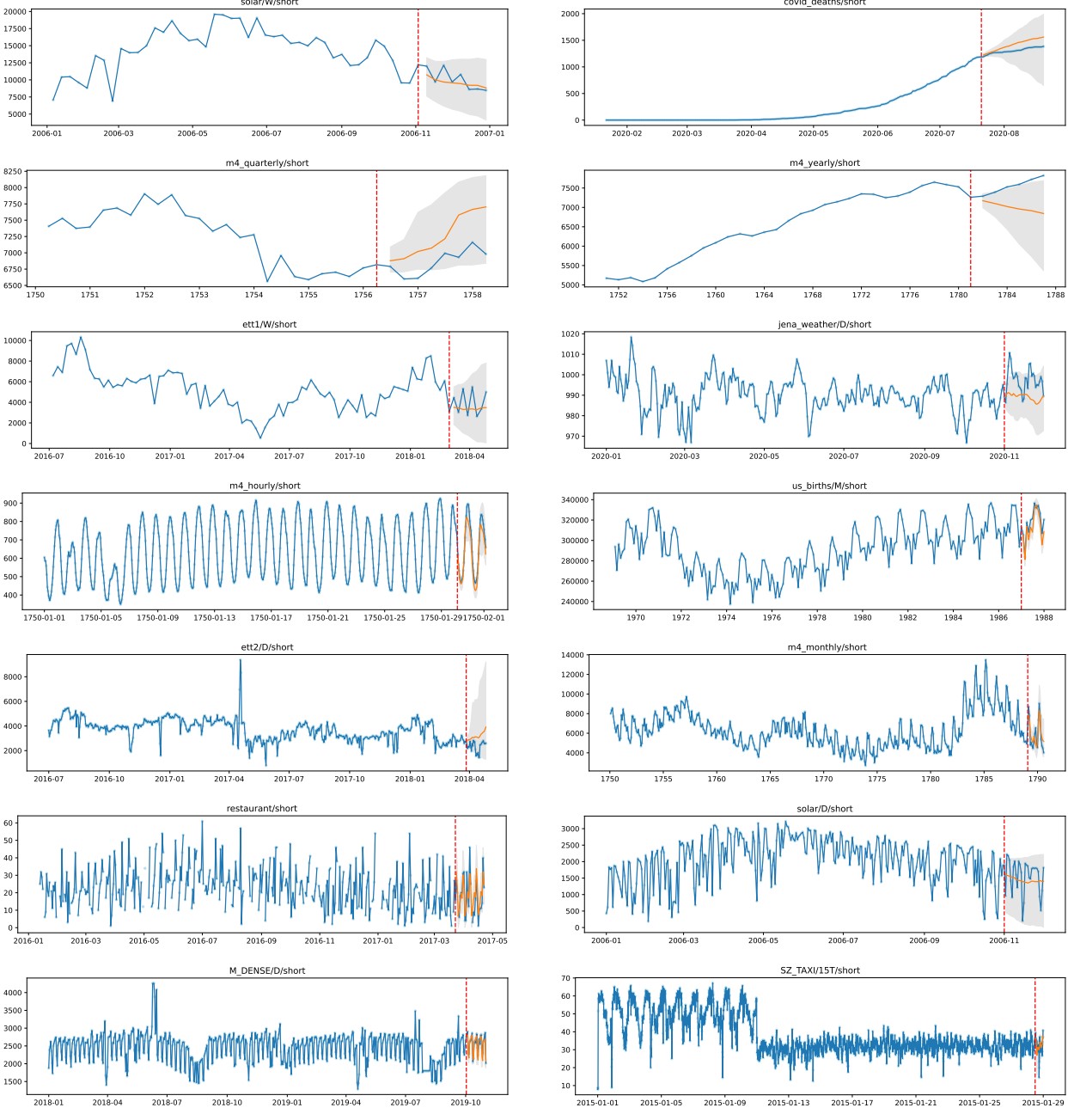

Figure 11: Visualization of the TabPFN-TS' predictions on some of the **short-term** benchmarking tasks.

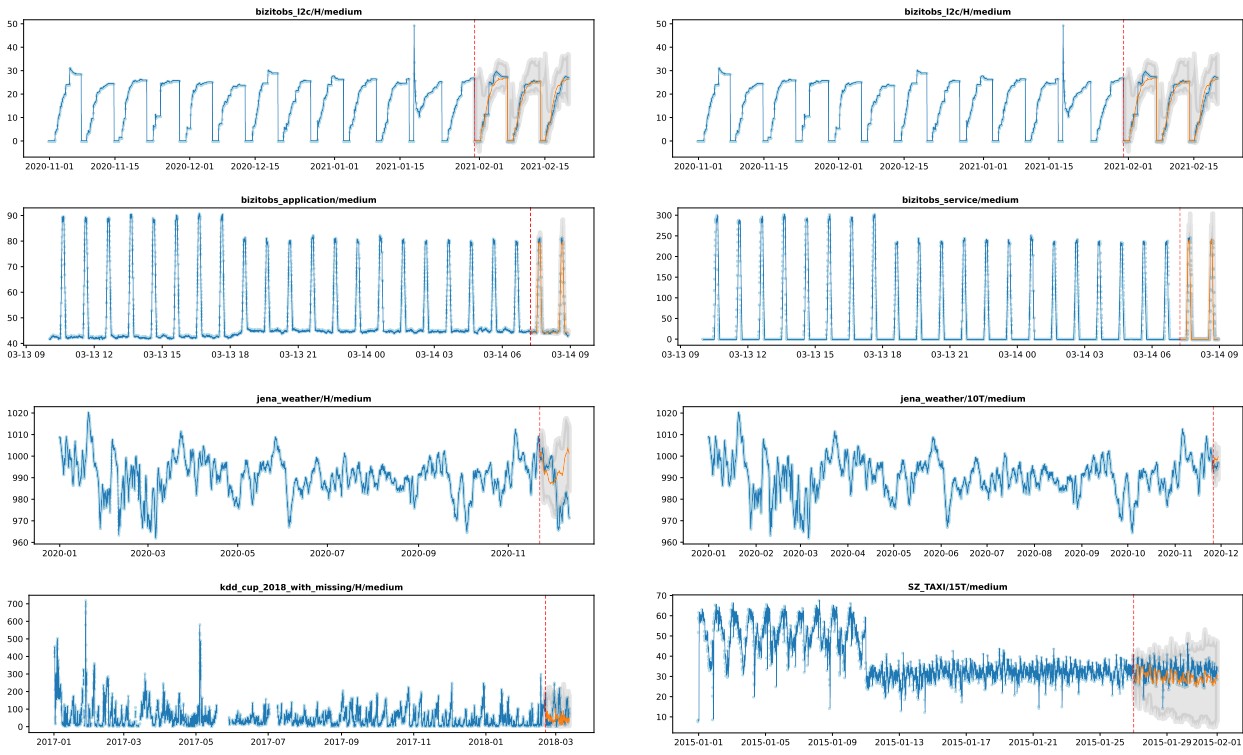

Figure 12: Visualization of the TabPFN-TS' predictions on some of the **medium-term** benchmarking tasks.

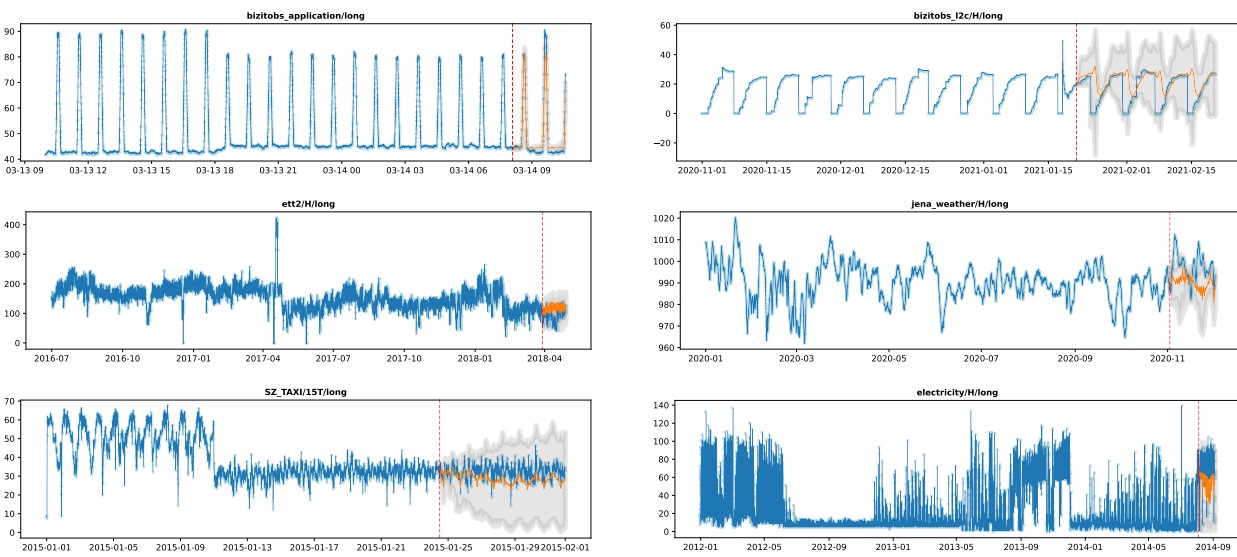

Figure 13: Visualization of the TabPFN-TS' predictions on some of the **long-term** benchmarking tasks.

Table 2: Probabilistic forecasting performance (WQL scores) of all models. Lower is better.

| Dataset | Freq. | Term | Tabular Foundation Model: TabPFN-TS | Time-Series Foundation Model: Chronos-Bolt Base | Chronos-Bolt Small | Chronos-Bolt Tiny | Moirai2 | Sundial | TiRex | TimesFM2.0-500M | Toto | Deep Learning Time-Series Model: DeepAR | PatchTST | TFT | Statistical Time-Series Model: AutoARIMA | AutoTheta | Seasonal Naive |
|---|---|---|---|---|---|---|---|---|---|---|---|---|---|---|---|---|---|
| bitbrains_fast_storage | 5T | long | 0.885 | 0.748 | 0.753 | 0.750 | 0.807 | 0.811 | 0.672 | 0.908 | **0.669** | 1.010 | 0.669 | 0.734 | 1.290 | 1.360 | 1.290 |
| | | medium | 0.949 | 0.755 | 0.867 | 0.814 | 0.694 | 0.728 | 0.638 | 0.881 | 0.629 | 0.990 | 0.642 | **0.610** | 1.270 | 1.450 | 1.270 |
| | | short | 0.662 | 0.454 | 0.435 | 0.420 | 0.427 | 0.462 | 0.380 | 0.447 | **0.371** | 0.493 | 0.471 | 0.451 | 1.210 | 1.210 | 1.210 |
| | H | short | 0.670 | 0.774 | 0.589 | 0.593 | 0.615 | 0.764 | 0.700 | 0.688 | 0.623 | 0.778 | **0.549** | 0.595 | 0.844 | 1.150 | 1.080 |
| bitbrains_rnd | 5T | long | 0.819 | 0.756 | 0.756 | 0.917 | **0.570** | 0.715 | 0.632 | 0.706 | 0.589 | 0.672 | 0.664 | 0.624 | 1.290 | 1.600 | 1.290 |
| | | medium | 0.819 | 0.605 | 0.792 | 0.697 | **0.596** | 0.730 | 0.604 | 0.727 | 0.628 | 0.647 | 0.620 | 0.628 | 1.260 | 1.470 | 1.260 |
| | | short | 0.608 | 0.438 | 0.453 | 0.482 | 0.404 | 0.433 | 0.404 | 0.461 | **0.399** | 0.557 | 0.474 | 0.486 | 1.100 | 0.741 | 1.100 |
| | H | short | 0.742 | 0.624 | 0.623 | 0.604 | 0.670 | 0.725 | 0.611 | 0.649 | 0.593 | **0.585** | 0.603 | 0.650 | 0.874 | 1.380 | 1.300 |
| bizitobs_application | 10S | long | 0.049 | 0.109 | 0.092 | 0.137 | 0.056 | 0.061 | 0.052 | 0.057 | 0.053 | 0.083 | 0.054 | 0.056 | 0.973 | **0.035** | 0.973 |
| | | medium | 0.041 | 0.104 | 0.085 | 0.115 | 0.037 | 0.046 | 0.038 | 0.033 | 0.034 | 0.053 | 0.047 | 0.047 | 0.042 | **0.024** | 0.042 |
| | | short | 0.015 | 0.054 | 0.035 | 0.070 | 0.013 | 0.016 | 0.011 | 0.014 | 0.012 | 0.064 | 0.022 | 0.090 | 0.035 | **0.010** | 0.035 |
| bizitobs_l2c | 5T | long | 0.306 | 0.738 | 0.790 | 0.722 | 0.300 | 0.310 | **0.269** | 0.748 | 0.533 | 0.719 | 0.324 | 0.472 | 0.674 | 0.632 | 0.674 |
| | | medium | 0.261 | 0.445 | 0.462 | 0.420 | 0.261 | **0.234** | 0.251 | 0.529 | 0.316 | 0.589 | 0.332 | 0.346 | 0.530 | 0.415 | 0.530 |
| | | short | 0.084 | 0.074 | 0.073 | 0.075 | 0.084 | **0.067** | 0.076 | 0.084 | 0.069 | 0.179 | 0.074 | 0.077 | 0.262 | 0.080 | 0.262 |
| | H | long | 0.292 | 0.278 | 0.295 | 0.306 | 0.321 | 0.325 | **0.268** | 0.728 | 0.369 | 0.338 | 0.291 | 0.286 | 0.787 | 0.819 | 1.820 |
| | | medium | **0.237** | 0.254 | 0.285 | 0.304 | 0.274 | 0.276 | 0.252 | 0.640 | 0.356 | 0.345 | 0.263 | 0.345 | 0.813 | 0.892 | 1.420 |
| | | short | 0.210 | **0.189** | 0.204 | 0.203 | 0.235 | 0.223 | 0.212 | 0.345 | 0.199 | 0.789 | 0.217 | 0.401 | 0.547 | 0.507 | 0.536 |
| bizitobs_service | 10S | long | 0.052 | 0.113 | 0.095 | 0.133 | 0.054 | 0.057 | 0.053 | 0.062 | **0.051** | 0.070 | 0.057 | 0.056 | 0.056 | 0.052 | 0.056 |
| | | medium | 0.041 | 0.096 | 0.081 | 0.113 | 0.034 | 0.044 | **0.023** | 0.038 | 0.027 | 0.044 | 0.045 | 0.044 | 0.049 | 0.027 | 0.049 |
| | | short | 0.019 | 0.051 | 0.031 | 0.065 | 0.014 | 0.016 | 0.012 | 0.015 | **0.011** | 0.032 | 0.025 | 0.025 | 0.040 | 0.013 | 0.040 |
| car_parts | M | short | 0.970 | 0.995 | 1.007 | 1.001 | 0.936 | 1.189 | 0.995 | 1.046 | 0.899 | 0.953 | 1.000 | **0.890** | 1.290 | 1.340 | 1.720 |
| covid_deaths | D | short | 0.041 | 0.047 | 0.043 | 0.067 | 0.028 | 0.131 | 0.032 | 0.062 | **0.027** | 0.177 | 0.067 | 0.037 | 0.030 | 0.095 | 0.125 |
| electricity | 15T | long | 0.081 | 0.084 | 0.086 | 0.092 | 0.083 | 0.082 | **0.078** | 0.083 | 0.086 | 0.155 | 0.081 | 0.084 | 0.129 | 0.401 | 0.129 |
| | | medium | 0.083 | 0.083 | 0.087 | 0.089 | 0.080 | 0.082 | **0.079** | 0.080 | 0.086 | 0.119 | 0.086 | 0.094 | 0.124 | 0.328 | 0.124 |
| | | short | 0.097 | 0.082 | 0.082 | 0.086 | **0.077** | 0.084 | 0.091 | 0.079 | 0.099 | 0.152 | 0.134 | 0.184 | 0.165 | 0.140 | 0.165 |
| | D | short | 0.063 | 0.055 | 0.058 | 0.057 | **0.054** | 0.064 | 0.054 | 0.060 | 0.059 | 0.078 | 0.083 | 0.084 | 0.083 | 0.088 | 0.122 |
| | H | long | 0.108 | 0.098 | 0.102 | 0.102 | 0.097 | 0.093 | 0.094 | 0.089 | **0.083** | 0.176 | 0.104 | 0.094 | 0.190 | 0.300 | 0.190 |
| | | medium | 0.088 | 0.081 | 0.084 | 0.092 | 0.080 | 0.080 | 0.079 | **0.073** | 0.075 | 0.454 | 0.081 | 0.091 | 0.156 | 0.254 | 0.156 |
| | | short | 0.072 | 0.064 | 0.067 | 0.072 | 0.065 | 0.069 | 0.064 | **0.054** | 0.069 | 0.094 | 0.079 | 0.089 | 0.109 | 0.177 | 0.109 |
| | W | short | 0.055 | 0.047 | 0.048 | **0.047** | 0.066 | 0.072 | 0.048 | 0.049 | 0.064 | 0.092 | 0.095 | 0.107 | 0.100 | 0.101 | 0.099 |
| ett1 | 15T | long | 0.259 | 0.298 | 0.296 | 0.332 | 0.268 | 0.253 | **0.234** | 0.283 | 0.251 | 2.220 | 0.247 | 0.280 | 0.396 | 1.390 | 0.396 |
| | | medium | 0.253 | 0.281 | 0.288 | 0.299 | 0.260 | 0.260 | **0.237** | 0.278 | 0.260 | 0.315 | 0.250 | 0.247 | 0.352 | 1.130 | 0.352 |
| | | short | 0.167 | **0.158** | 0.169 | 0.179 | 0.160 | 0.177 | 0.161 | 0.168 | 0.162 | 0.320 | 0.191 | 0.245 | 0.241 | 0.410 | 0.241 |
| | D | short | 0.298 | 0.287 | 0.283 | 0.301 | 0.287 | 0.373 | **0.277** | 0.281 | 0.284 | 0.293 | 0.304 | 0.330 | 0.279 | 0.341 | 0.515 |
| | H | long | 0.295 | 0.311 | 0.337 | 0.317 | 0.323 | 0.283 | **0.258** | 0.310 | 0.267 | 0.469 | 0.297 | 0.313 | 0.430 | 1.940 | 0.616 |
| | | medium | 0.283 | 0.303 | 0.295 | 0.280 | 0.287 | 0.269 | **0.252** | 0.282 | 0.254 | 0.535 | 0.273 | 0.316 | 0.384 | 1.650 | 0.540 |
| | | short | 0.194 | 0.181 | 0.189 | 0.195 | 0.185 | 0.190 | **0.176** | 0.192 | 0.194 | 0.233 | 0.190 | 0.199 | 0.223 | 0.668 | 0.250 |
| | W | short | 0.284 | 0.296 | 0.293 | 0.275 | **0.249** | 0.404 | 0.278 | 0.272 | 0.263 | 0.686 | 0.323 | 0.406 | 0.305 | 0.319 | 0.338 |
| ett2 | 15T | long | 0.101 | 0.111 | 0.118 | 0.119 | 0.102 | 0.098 | 0.092 | 0.106 | **0.088** | 0.304 | 0.098 | 0.109 | 0.165 | 0.169 | 0.165 |
| | | medium | 0.100 | 0.110 | 0.119 | 0.113 | 0.098 | 0.096 | **0.089** | 0.105 | 0.093 | 0.258 | 0.094 | 0.104 | 0.143 | 0.150 | 0.143 |
| | | short | 0.073 | 0.067 | 0.070 | 0.070 | 0.066 | 0.069 | 0.066 | **0.065** | 0.068 | 0.378 | 0.076 | 0.081 | 0.096 | 0.077 | 0.096 |
| | D | short | 0.126 | 0.094 | **0.091** | 0.095 | 0.093 | 0.103 | 0.094 | 0.108 | 0.111 | 0.207 | 0.131 | 0.096 | 0.125 | 0.164 | 0.205 |
| | H | long | 0.139 | 0.117 | 0.121 | 0.124 | 0.109 | 0.117 | 0.114 | 0.125 | **0.108** | 0.196 | 0.130 | 0.138 | 0.272 | 0.336 | 0.287 |
| | | medium | 0.121 | 0.115 | 0.118 | 0.116 | 0.111 | 0.114 | 0.107 | 0.110 | **0.102** | 0.281 | 0.125 | 0.122 | 0.245 | 0.284 | 0.241 |
| | | short | 0.073 | **0.063** | 0.065 | 0.065 | 0.064 | 0.072 | 0.064 | 0.066 | 0.065 | 0.122 | 0.074 | 0.078 | 0.089 | 0.102 | 0.094 |
| | W | short | 0.099 | 0.088 | 0.094 | 0.095 | **0.085** | 0.098 | 0.087 | 0.110 | 0.106 | 0.728 | 0.142 | 0.160 | 0.136 | 0.160 | 0.169 |
| hierarchical_sales | D | short | 0.592 | 0.576 | 0.582 | 0.581 | 0.577 | 0.649 | **0.570** | 0.576 | 0.570 | 0.600 | 0.590 | 0.600 | 0.735 | 0.967 | 2.360 |
| | W | short | 0.345 | 0.353 | 0.354 | 0.353 | 0.352 | 0.390 | 0.348 | **0.330** | 0.356 | 0.379 | 0.358 | 0.382 | 0.485 | 0.474 | 1.030 |
| hospital | M | short | 0.054 | 0.057 | 0.058 | 0.059 | 0.052 | 0.061 | 0.051 | **0.050** | 0.052 | 0.062 | 0.064 | 0.058 | 0.060 | 0.055 | 0.062 |
| jena_weather | 10T | long | 0.053 | 0.064 | 0.063 | 0.079 | 0.060 | 0.056 | 0.049 | **0.035** | 0.050 | 0.143 | 0.066 | 0.052 | 0.304 | 0.424 | 0.304 |
| | | medium | 0.054 | 0.057 | 0.060 | 0.068 | 0.059 | 0.054 | 0.048 | **0.031** | 0.049 | 0.073 | 0.065 | 0.052 | 0.277 | 0.350 | 0.277 |
| | | short | 0.034 | 0.033 | 0.037 | 0.042 | 0.036 | 0.031 | 0.027 | **0.016** | 0.027 | 0.063 | 0.064 | 0.069 | 0.155 | 0.130 | 0.155 |
| | D | short | 0.047 | 0.045 | 0.047 | 0.047 | **0.043** | 0.048 | 0.044 | 0.058 | 0.051 | 0.062 | 0.053 | 0.069 | 0.080 | 0.082 | 0.297 |
| | H | long | 0.103 | 0.062 | 0.068 | 0.066 | 0.058 | 0.066 | 0.059 | 0.068 | **0.057** | 0.197 | 0.076 | 0.090 | 0.230 | 1.290 | 0.598 |
| | | medium | 0.058 | 0.054 | 0.058 | 0.058 | 0.055 | 0.058 | 0.053 | 0.066 | **0.053** | 0.078 | 0.069 | 0.073 | 0.211 | 0.832 | 0.486 |
| | | short | 0.042 | 0.042 | 0.043 | 0.042 | 0.042 | 0.050 | **0.041** | 0.045 | 0.042 | 0.699 | 0.050 | 0.048 | 0.143 | 0.296 | 0.173 |
| kdd_cup_2018 | D | short | **0.362** | 0.372 | 0.373 | 0.389 | 0.389 | 0.396 | 0.376 | 0.378 | 0.387 | 0.383 | 0.401 | 0.380 | 0.393 | 0.459 | 0.888 |
| | H | long | 0.478 | **0.300** | 0.419 | 0.472 | 0.518 | 0.375 | 0.441 | 0.518 | 0.457 | 1.090 | 0.477 | 0.503 | 1.050 | 0.970 | 1.250 |
| | | medium | 0.450 | **0.301** | 0.364 | 0.416 | 0.495 | 0.377 | 0.421 | 0.466 | 0.441 | 0.442 | 0.442 | 0.472 | 0.851 | 0.791 | 0.949 |
| | | short | 0.418 | **0.246** | 0.267 | 0.313 | 0.426 | 0.351 | 0.378 | 0.376 | 0.403 | 0.517 | 0.457 | 0.467 | 0.559 | 0.531 | 0.559 |
| loop_seattle | 5T | long | 0.090 | 0.129 | 0.125 | 0.121 | 0.087 | 0.084 | 0.084 | 0.114 | **0.077** | 0.184 | 0.095 | 0.088 | 0.137 | 0.231 | 0.137 |
| | | medium | 0.087 | 0.116 | 0.119 | 0.116 | 0.080 | 0.077 | 0.078 | 0.110 | **0.072** | 0.118 | 0.095 | 0.092 | 0.123 | 0.240 | 0.123 |
| | | short | 0.053 | 0.055 | 0.055 | 0.055 | **0.046** | 0.050 | 0.048 | 0.051 | 0.048 | 0.072 | 0.066 | 0.065 | 0.081 | 0.082 | 0.081 |
| | D | short | 0.043 | 0.044 | 0.045 | 0.046 | 0.043 | 0.047 | 0.042 | **0.041** | 0.044 | 0.052 | 0.046 | 0.048 | 0.078 | 0.072 | 0.131 |
| | H | long | 0.063 | 0.076 | 0.082 | 0.087 | 0.066 | 0.072 | **0.060** | 0.066 | 0.065 | 0.068 | 0.069 | 0.068 | 0.193 | 0.468 | 0.245 |
| | | medium | 0.067 | 0.076 | 0.082 | 0.087 | 0.069 | 0.075 | **0.064** | 0.067 | 0.064 | 0.072 | 0.071 | 0.069 | 0.154 | 0.390 | 0.206 |
| | | short | 0.063 | 0.065 | 0.066 | 0.071 | 0.062 | 0.067 | 0.058 | 0.059 | 0.063 | 0.066 | 0.076 | 0.073 | 0.108 | 0.165 | 0.108 |
| m4_daily | D | short | 0.023 | 0.021 | 0.021 | 0.021 | **0.020** | 0.027 | 0.021 | 0.021 | 0.022 | 0.030 | 0.023 | 0.023 | 0.023 | 0.024 | 0.026 |
| m4_hourly | H | short | 0.030 | 0.025 | 0.020 | 0.021 | 0.024 | 0.023 | 0.020 | **0.011** | 0.035 | 0.133 | 0.039 | 0.040 | 0.034 | 0.041 | 0.040 |
| m4_monthly | M | short | 0.094 | 0.094 | 0.094 | 0.095 | 0.095 | 0.116 | 0.091 | **0.067** | 0.097 | 0.184 | 0.102 | 0.113 | 0.098 | 0.098 | 0.126 |
| m4_quarterly | Q | short | 0.078 | 0.077 | 0.078 | 0.079 | 0.075 | 0.093 | 0.073 | **0.062** | 0.078 | 0.083 | 0.083 | 0.083 | 0.082 | 0.079 | 0.099 |
| m4_weekly | W | short | 0.037 | 0.038 | 0.038 | 0.041 | 0.040 | 0.043 | **0.037** | 0.042 | 0.049 | 0.062 | 0.040 | 0.049 | 0.050 | 0.053 | 0.073 |
| m4_yearly | A | short | 0.118 | 0.121 | 0.128 | 0.129 | 0.116 | 0.160 | 0.116 | **0.091** | 0.122 | 0.113 | 0.117 | 0.110 | 0.130 | 0.115 | 0.138 |
| m_dense | D | short | 0.061 | 0.069 | 0.072 | 0.082 | 0.070 | 0.067 | 0.068 | **0.060** | 0.075 | 0.076 | 0.070 | 0.077 | 0.135 | 0.126 | 0.294 |
| | H | long | 0.165 | 0.170 | 0.146 | 0.198 | 0.120 | 0.130 | 0.115 | 0.127 | 0.128 | 0.130 | 0.120 | **0.115** | 0.270 | 1.430 | 0.552 |
| | | medium | 0.160 | 0.157 | 0.134 | 0.155 | 0.120 | 0.128 | 0.116 | 0.127 | 0.121 | 0.118 | 0.127 | **0.114** | 0.255 | 1.210 | 0.479 |
| | | short | 0.155 | **0.125** | 0.133 | 0.140 | 0.132 | 0.133 | 0.128 | 0.139 | 0.148 | 0.128 | 0.173 | 0.139 | 0.281 | 0.549 | 0.281 |
| restaurant | D | short | 0.263 | 0.264 | 0.264 | 0.276 | 0.260 | 0.286 | **0.255** | 0.261 | 0.297 | 0.270 | 0.262 | 0.284 | 0.362 | 0.329 | 0.907 |
| saugeen | D | short | 0.373 | 0.338 | 0.354 | 0.339 | **0.328** | 0.379 | 0.358 | 0.408 | 0.353 | 0.572 | 0.408 | 0.419 | 0.564 | 0.669 | 0.754 |
| | M | short | **0.276** | 0.296 | 0.293 | 0.288 | 0.291 | 0.332 | 0.297 | 0.342 | 0.299 | 0.689 | 0.372 | 0.340 | 0.326 | 0.373 | 0.445 |
| | W | short | 0.395 | 0.363 | 0.372 | 0.364 | 0.419 | 0.406 | **0.350** | 0.601 | 0.390 | 0.397 | 0.484 | 0.491 | 0.549 | 0.734 | 0.855 |
| solar | 10T | long | 0.331 | 0.443 | 0.497 | 0.534 | 0.408 | 0.365 | **0.321** | 0.498 | 0.352 | 0.549 | 0.339 | 0.379 | 0.786 | 6.640 | 0.786 |
| | | medium | **0.326** | 0.436 | 0.453 | 0.495 | 0.412 | 0.373 | 0.334 | 0.516 | 0.353 | 0.485 | 0.356 | 0.362 | 0.771 | 5.670 | 0.771 |
| | | short | 0.458 | 0.511 | 0.498 | 0.488 | **0.350** | 0.444 | 0.541 | 0.804 | 0.541 | 0.933 | 1.370 | 0.618 | 0.860 | 2.360 | 0.860 |
| | D | short | **0.269** | 0.287 | 0.286 | 0.282 | 0.303 | 0.324 | 0.273 | 0.278 | 0.290 | 0.682 | 0.287 | 0.277 | 0.282 | 0.286 | 0.757 |
| | H | long | 0.351 | 0.405 | 0.373 | 0.354 | 0.332 | 0.293 | **0.287** | 0.493 | 0.331 | 0.381 | 0.353 | 0.401 | 0.607 | 7.320 | 1.470 |
| | | medium | 0.313 | 0.368 | 0.356 | 0.342 | 0.333 | 0.309 | **0.285** | 0.376 | 0.331 | 0.352 | 0.344 | 0.330 | 0.557 | 6.130 | 1.270 |
| | | short | 0.336 | 0.298 | 0.303 | 0.313 | 0.342 | 0.329 | **0.273** | 0.406 | 0.328 | 0.389 | 0.340 | 0.367 | 0.628 | 2.330 | 0.628 |
| | W | short | 0.120 | 0.133 | 0.136 | 0.132 | 0.163 | 0.148 | 0.144 | 0.171 | 0.186 | 0.242 | 0.162 | **0.114** | 0.152 | 0.155 | 0.236 |
| sz_taxi | 15T | long | 0.242 | 0.248 | 0.245 | 0.240 | 0.217 | 0.221 | **0.198** | 0.227 | 0.202 | 0.286 | 0.281 | 0.241 | 0.398 | 0.629 | 0.554 |
| | | medium | 0.230 | 0.244 | 0.246 | 0.240 | 0.214 | 0.228 | **0.202** | 0.229 | 0.205 | 0.210 | 0.220 | 0.206 | 0.351 | 0.529 | 0.454 |
| | | short | 0.209 | 0.202 | 0.203 | 0.203 | 0.201 | 0.223 | 0.200 | **0.199** | 0.203 | 0.219 | 0.207 | 0.222 | 0.309 | 0.288 | 0.309 |
| | D | short | 0.140 | 0.136 | 0.137 | 0.137 | 0.136 | 0.154 | 0.135 | **0.135** | 0.137 | 0.139 | 0.144 | 0.144 | 0.170 | 0.232 | 0.229 |
| temperature_rain | D | short | 0.569 | **0.538** | 0.544 | 0.548 | 0.561 | 0.620 | 0.547 | 0.586 | 0.560 | 0.682 | 0.644 | 0.592 | 0.694 | 0.761 | 1.630 |
| us_births | D | short | **0.016** | 0.026 | 0.028 | 0.037 | 0.020 | 0.022 | 0.023 | 0.019 | 0.026 | 0.028 | 0.025 | 0.016 | 0.074 | 0.075 | 0.144 |
| | M | short | 0.015 | 0.019 | 0.016 | 0.017 | 0.017 | 0.028 | 0.012 | 0.011 | 0.013 | 0.016 | 0.017 | 0.021 | **0.010** | 0.019 | 0.017 |
| | W | short | **0.011** | 0.013 | 0.013 | 0.014 | 0.011 | 0.017 | 0.012 | 0.013 | 0.014 | 0.017 | 0.015 | 0.019 | 0.018 | 0.018 | 0.022 |

Table 3: Point forecasting performance (MASE scores) of all models. Lower is better.

| Dataset | Freq. | Term | Tabular Foundation Model | Time-Series Foundation Model | | | | | | | | Deep Learning Time-Series Model | | | Statistical Time-Series Model | | |
|---|---|---|---|---|---|---|---|---|---|---|---|---|---|---|---|---|---|
| | | | TabPFN-TS | Chronos-Bolt Base | Chronos-Bolt Small | Chronos-Bolt Tiny | Moirai2 | Sundial | TiRex | TimesFM2.0-500M | Toto | DeepAR | PatchTST | TFT | AutoARIMA | AutoTheta | Seasonal Naive |
| bitbrains_fast_storage | 5T | long | 0.885 | 0.748 | 0.753 | 0.750 | 0.807 | 0.811 | 0.672 | 0.908 | **0.669** | 1.010 | 0.669 | 0.734 | 1.290 | 1.360 | 1.290 |
| | | medium | 0.949 | 0.755 | 0.867 | 0.814 | 0.694 | 0.728 | 0.638 | 0.881 | 0.629 | 0.990 | 0.642 | **0.610** | 1.270 | 1.450 | 1.270 |
| | | short | 0.662 | 0.454 | 0.435 | 0.420 | 0.427 | 0.462 | 0.380 | 0.447 | **0.371** | 0.493 | 0.471 | 0.451 | 1.210 | 0.731 | 1.210 |
| | H | short | 0.670 | 0.774 | 0.589 | 0.593 | 0.615 | 0.764 | 0.700 | 0.688 | 0.623 | 0.778 | **0.549** | 0.595 | 0.844 | 1.150 | 1.080 |
| bitbrains_rnd | 5T | long | 0.819 | 0.756 | 0.756 | 0.917 | **0.570** | 0.715 | 0.632 | 0.706 | 0.589 | 0.672 | 0.664 | 0.624 | 1.290 | 1.600 | 1.290 |
| | | medium | 0.819 | 0.605 | 0.792 | 0.697 | **0.596** | 0.730 | 0.604 | 0.727 | 0.628 | 0.647 | 0.620 | 0.628 | 1.260 | 1.470 | 1.260 |
| | | short | 0.608 | 0.438 | 0.453 | 0.482 | 0.404 | 0.433 | 0.404 | 0.461 | **0.399** | 0.557 | 0.474 | 0.486 | 1.100 | 0.741 | 1.100 |
| | H | short | 0.742 | 0.624 | 0.623 | 0.604 | 0.670 | 0.725 | 0.611 | 0.649 | 0.593 | **0.585** | 0.603 | 0.650 | 0.874 | 1.380 | 1.300 |
| bizitobs_application | 10S | long | 0.049 | 0.109 | 0.092 | 0.137 | 0.056 | 0.061 | 0.052 | 0.057 | 0.053 | 0.083 | 0.054 | 0.056 | 0.973 | **0.035** | 0.973 |
| | | medium | 0.041 | 0.104 | 0.085 | 0.115 | 0.037 | 0.046 | 0.038 | 0.033 | 0.034 | 0.053 | 0.047 | 0.047 | 0.042 | **0.024** | 0.042 |
| | | short | 0.015 | 0.054 | 0.035 | 0.070 | 0.013 | 0.016 | 0.011 | 0.014 | 0.012 | 0.064 | 0.022 | 0.090 | 0.035 | **0.010** | 0.035 |
| bizitobs_l2c | 5T | long | 0.306 | 0.738 | 0.790 | 0.722 | 0.300 | 0.310 | **0.269** | 0.748 | 0.533 | 0.719 | 0.324 | 0.472 | 0.674 | 0.632 | 0.674 |
| | | medium | 0.261 | 0.445 | 0.462 | 0.420 | 0.261 | **0.234** | 0.251 | 0.529 | 0.316 | 0.589 | 0.332 | 0.346 | 0.530 | 0.415 | 0.530 |
| | | short | 0.084 | 0.074 | 0.073 | 0.075 | 0.084 | **0.067** | 0.076 | 0.084 | 0.069 | 0.179 | 0.074 | 0.077 | 0.262 | 0.080 | 0.262 |
| | H | long | 0.292 | 0.278 | 0.295 | 0.306 | 0.321 | 0.325 | **0.268** | 0.728 | 0.369 | 0.338 | 0.291 | 0.286 | 0.787 | 0.819 | 1.820 |
| | | medium | **0.237** | 0.254 | 0.285 | 0.304 | 0.274 | 0.276 | 0.252 | 0.640 | 0.356 | 0.345 | 0.263 | 0.345 | 0.813 | 0.892 | 1.420 |
| | | short | 0.210 | **0.189** | 0.204 | 0.203 | 0.235 | 0.223 | 0.212 | 0.345 | 0.199 | 0.789 | 0.217 | 0.401 | 0.547 | 0.507 | 0.536 |
| bizitobs_service | 10S | long | 0.052 | 0.113 | 0.095 | 0.133 | 0.054 | 0.057 | 0.053 | 0.062 | **0.051** | 0.070 | 0.057 | 0.056 | 0.056 | 0.052 | 0.056 |
| | | medium | 0.041 | 0.096 | 0.081 | 0.113 | 0.034 | 0.044 | **0.023** | 0.038 | 0.027 | 0.044 | 0.045 | 0.044 | 0.049 | 0.027 | 0.049 |
| | | short | 0.019 | 0.051 | 0.031 | 0.065 | 0.014 | 0.016 | 0.012 | 0.015 | **0.011** | 0.032 | 0.025 | 0.025 | 0.040 | 0.013 | 0.040 |
| car_parts | M | short | 0.970 | 0.995 | 1.007 | 1.001 | 0.936 | 1.189 | 0.995 | 1.046 | 0.899 | 0.953 | 1.000 | **0.890** | 1.290 | 1.340 | 1.720 |
| covid_deaths | D | short | 0.041 | 0.047 | 0.043 | 0.067 | 0.028 | 0.131 | 0.032 | 0.062 | **0.027** | 0.177 | 0.067 | 0.037 | 0.030 | 0.095 | 0.125 |
| electricity | 15T | long | 0.081 | 0.084 | 0.086 | 0.092 | 0.083 | 0.082 | **0.078** | 0.083 | 0.086 | 0.155 | 0.081 | 0.084 | 0.129 | 0.401 | 0.129 |
| | | medium | 0.083 | 0.083 | 0.087 | 0.089 | 0.080 | 0.082 | **0.079** | 0.080 | 0.086 | 0.119 | 0.086 | 0.094 | 0.124 | 0.328 | 0.124 |
| | | short | 0.097 | 0.082 | 0.082 | 0.086 | **0.077** | 0.084 | 0.091 | 0.079 | 0.099 | 0.152 | 0.134 | 0.184 | 0.165 | 0.140 | 0.165 |
| | D | short | 0.063 | 0.055 | 0.058 | 0.057 | **0.054** | 0.064 | 0.054 | 0.060 | 0.059 | 0.078 | 0.083 | 0.084 | 0.083 | 0.088 | 0.122 |
| | H | long | 0.108 | 0.098 | 0.102 | 0.102 | 0.097 | 0.093 | 0.094 | 0.089 | **0.083** | 0.176 | 0.104 | 0.094 | 0.190 | 0.300 | 0.190 |
| | | medium | 0.088 | 0.081 | 0.084 | 0.092 | 0.080 | 0.080 | 0.079 | **0.073** | 0.075 | 0.454 | 0.081 | 0.091 | 0.156 | 0.254 | 0.156 |
| | | short | 0.072 | 0.064 | 0.067 | 0.072 | 0.065 | 0.069 | 0.064 | **0.054** | 0.069 | 0.094 | 0.079 | 0.089 | 0.109 | 0.177 | 0.109 |
| | W | short | 0.055 | 0.047 | 0.048 | **0.047** | 0.066 | 0.072 | 0.048 | 0.049 | 0.064 | 0.092 | 0.095 | 0.107 | 0.100 | 0.101 | 0.099 |
| ett1 | 15T | long | 0.259 | 0.298 | 0.296 | 0.332 | 0.268 | 0.253 | **0.234** | 0.283 | 0.251 | 2.220 | 0.247 | 0.280 | 0.396 | 1.390 | 0.396 |
| | | medium | 0.253 | 0.281 | 0.288 | 0.299 | 0.260 | 0.260 | **0.237** | 0.278 | 0.260 | 0.315 | 0.250 | 0.247 | 0.352 | 1.130 | 0.352 |
| | | short | 0.167 | **0.158** | 0.169 | 0.179 | 0.160 | 0.177 | 0.161 | 0.168 | 0.162 | 0.320 | 0.191 | 0.245 | 0.241 | 0.410 | 0.241 |
| | D | short | 0.298 | 0.287 | 0.283 | 0.301 | 0.287 | 0.373 | **0.277** | 0.284 | 0.284 | 0.293 | 0.304 | 0.330 | 0.279 | 0.341 | 0.515 |
| | H | long | 0.295 | 0.311 | 0.337 | 0.317 | 0.323 | 0.283 | **0.258** | 0.310 | 0.267 | 0.469 | 0.297 | 0.313 | 0.430 | 1.940 | 0.616 |
| | | medium | 0.283 | 0.303 | 0.295 | 0.280 | 0.287 | 0.269 | **0.252** | 0.282 | 0.254 | 0.535 | 0.273 | 0.316 | 0.384 | 1.650 | 0.540 |
| | | short | 0.194 | 0.181 | 0.189 | 0.195 | 0.185 | 0.190 | **0.176** | 0.192 | 0.194 | 0.233 | 0.190 | 0.199 | 0.223 | 0.668 | 0.250 |
| | W | short | 0.284 | 0.296 | 0.293 | 0.275 | **0.249** | 0.404 | 0.278 | 0.272 | 0.263 | 0.686 | 0.323 | 0.406 | 0.305 | 0.319 | 0.338 |
| ett2 | 15T | long | 0.101 | 0.111 | 0.118 | 0.119 | 0.102 | 0.098 | 0.092 | 0.106 | **0.088** | 0.304 | 0.098 | 0.109 | 0.165 | 0.169 | 0.165 |
| | | medium | 0.100 | 0.110 | 0.119 | 0.113 | 0.098 | 0.096 | **0.089** | 0.105 | 0.093 | 0.258 | 0.094 | 0.104 | 0.143 | 0.150 | 0.143 |
| | | short | 0.073 | 0.067 | 0.070 | 0.070 | 0.066 | 0.069 | 0.066 | **0.065** | 0.068 | 0.378 | 0.076 | 0.081 | 0.096 | 0.077 | 0.096 |
| | D | short | 0.126 | 0.094 | **0.091** | 0.095 | 0.093 | 0.103 | 0.094 | 0.108 | 0.111 | 0.207 | 0.131 | 0.096 | 0.125 | 0.164 | 0.205 |
| | H | long | 0.139 | 0.117 | 0.121 | 0.124 | 0.109 | 0.117 | 0.114 | 0.125 | **0.108** | 0.196 | 0.130 | 0.138 | 0.272 | 0.336 | 0.287 |
| | | medium | 0.121 | 0.115 | 0.118 | 0.116 | 0.111 | 0.114 | 0.107 | 0.110 | **0.102** | 0.281 | 0.125 | 0.122 | 0.245 | 0.284 | 0.241 |
| | | short | 0.073 | **0.063** | 0.065 | 0.065 | 0.064 | 0.072 | 0.064 | 0.066 | 0.065 | 0.122 | 0.074 | 0.078 | 0.089 | 0.102 | 0.094 |
| | W | short | 0.099 | 0.088 | 0.094 | 0.095 | **0.085** | 0.098 | 0.087 | 0.110 | 0.106 | 0.728 | 0.142 | 0.160 | 0.136 | 0.160 | 0.169 |
| hierarchical_sales | D | short | 0.592 | 0.576 | 0.582 | 0.581 | 0.577 | 0.649 | **0.570** | 0.576 | 0.570 | 0.600 | 0.590 | 0.600 | 0.735 | 0.967 | 2.360 |
| | W | short | 0.345 | 0.353 | 0.354 | 0.353 | 0.352 | 0.390 | 0.348 | **0.330** | 0.356 | 0.379 | 0.358 | 0.382 | 0.485 | 0.474 | 1.030 |
| hospital | M | short | 0.054 | 0.057 | 0.058 | 0.059 | 0.052 | 0.061 | 0.051 | **0.050** | 0.052 | 0.062 | 0.064 | 0.058 | 0.060 | 0.055 | 0.062 |
| jena_weather | 10T | long | 0.053 | 0.064 | 0.063 | 0.079 | 0.060 | 0.056 | 0.049 | **0.035** | 0.050 | 0.143 | 0.066 | 0.052 | 0.304 | 0.424 | 0.304 |
| | | medium | 0.054 | 0.057 | 0.060 | 0.068 | 0.059 | 0.054 | 0.048 | **0.031** | 0.049 | 0.073 | 0.065 | 0.052 | 0.277 | 0.350 | 0.277 |
| | | short | 0.034 | 0.033 | 0.037 | 0.042 | 0.036 | 0.031 | 0.027 | **0.016** | 0.027 | 0.063 | 0.064 | 0.069 | 0.155 | 0.130 | 0.155 |
| | D | short | 0.047 | 0.045 | 0.047 | 0.047 | **0.043** | 0.048 | 0.044 | 0.058 | 0.051 | 0.062 | 0.053 | 0.069 | 0.080 | 0.082 | 0.297 |
| | H | long | 0.103 | 0.062 | 0.068 | 0.068 | 0.058 | 0.066 | 0.059 | 0.068 | **0.057** | 0.197 | 0.076 | 0.090 | 0.230 | 1.290 | 0.598 |
| | | medium | 0.058 | 0.054 | 0.058 | 0.058 | 0.055 | 0.058 | 0.053 | 0.066 | **0.053** | 0.078 | 0.069 | 0.073 | 0.211 | 0.832 | 0.486 |
| | | short | 0.042 | 0.042 | 0.043 | 0.042 | 0.042 | 0.050 | **0.041** | 0.045 | 0.042 | 0.699 | 0.050 | 0.048 | 0.143 | 0.296 | 0.173 |
| kdd_cup_2018 | D | short | **0.362** | 0.372 | 0.373 | 0.365 | 0.389 | 0.396 | 0.376 | 0.378 | 0.387 | 0.383 | 0.401 | 0.380 | 0.393 | 0.459 | 0.888 |
| | H | long | 0.478 | **0.300** | 0.419 | 0.472 | 0.518 | 0.375 | 0.441 | 0.518 | 0.457 | 1.090 | 0.477 | 0.503 | 1.050 | 0.970 | 1.250 |
| | | medium | 0.450 | **0.301** | 0.364 | 0.416 | 0.495 | 0.377 | 0.421 | 0.466 | 0.441 | 0.442 | 0.442 | 0.472 | 0.851 | 0.791 | 0.949 |
| | | short | 0.418 | **0.246** | 0.267 | 0.313 | 0.426 | 0.351 | 0.378 | 0.376 | 0.403 | 0.517 | 0.457 | 0.467 | 0.559 | 0.531 | 0.559 |
| loop_seattle | 5T | long | 0.090 | 0.129 | 0.125 | 0.121 | 0.087 | 0.084 | 0.084 | 0.114 | **0.077** | 0.184 | 0.095 | 0.088 | 0.137 | 0.231 | 0.137 |
| | | medium | 0.087 | 0.116 | 0.119 | 0.116 | 0.080 | 0.077 | 0.078 | 0.110 | **0.072** | 0.118 | 0.095 | 0.092 | 0.123 | 0.240 | 0.123 |
| | | short | 0.053 | 0.055 | 0.055 | 0.055 | **0.046** | 0.050 | 0.048 | 0.051 | 0.048 | 0.072 | 0.066 | 0.065 | 0.081 | 0.082 | 0.081 |
| | D | short | 0.043 | 0.044 | 0.045 | 0.046 | 0.043 | 0.047 | 0.042 | **0.041** | 0.044 | 0.052 | 0.046 | 0.048 | 0.078 | 0.072 | 0.131 |
| | H | long | 0.063 | 0.076 | 0.082 | 0.087 | 0.066 | 0.072 | **0.060** | 0.066 | 0.065 | 0.068 | 0.069 | 0.068 | 0.193 | 0.468 | 0.245 |
| | | medium | 0.067 | 0.076 | 0.082 | 0.087 | 0.069 | 0.075 | **0.064** | 0.067 | 0.064 | 0.072 | 0.071 | 0.069 | 0.154 | 0.390 | 0.206 |
| | | short | 0.063 | 0.065 | 0.066 | 0.071 | 0.062 | 0.067 | **0.058** | 0.059 | 0.063 | 0.066 | 0.076 | 0.073 | 0.108 | 0.165 | 0.108 |
| m4_daily | D | short | 0.023 | 0.021 | 0.021 | 0.021 | **0.020** | 0.027 | 0.021 | 0.021 | 0.022 | 0.030 | 0.023 | 0.023 | 0.023 | 0.024 | 0.026 |
| m4_hourly | H | short | 0.030 | 0.025 | 0.020 | 0.021 | 0.024 | 0.023 | 0.020 | **0.011** | 0.035 | 0.133 | 0.039 | 0.040 | 0.034 | 0.041 | 0.040 |
| m4_monthly | M | short | 0.094 | 0.094 | 0.094 | 0.095 | 0.095 | 0.116 | 0.091 | **0.067** | 0.097 | 0.184 | 0.102 | 0.113 | 0.098 | 0.098 | 0.126 |
| m4_quarterly | Q | short | 0.078 | 0.077 | 0.078 | 0.079 | 0.075 | 0.093 | 0.073 | **0.062** | 0.078 | 0.083 | 0.083 | 0.083 | 0.082 | 0.079 | 0.099 |
| m4_weekly | W | short | 0.037 | 0.038 | 0.038 | 0.041 | 0.040 | 0.043 | **0.037** | 0.042 | 0.049 | 0.062 | 0.040 | 0.049 | 0.050 | 0.053 | 0.073 |
| m4_yearly | A | short | 0.118 | 0.121 | 0.128 | 0.129 | 0.116 | 0.160 | 0.116 | **0.091** | 0.122 | 0.113 | 0.117 | 0.110 | 0.130 | 0.115 | 0.138 |
| m_dense | D | short | 0.061 | 0.069 | 0.072 | 0.082 | 0.070 | 0.067 | 0.068 | **0.060** | 0.075 | 0.076 | 0.070 | 0.077 | 0.135 | 0.126 | 0.294 |
| | H | long | 0.165 | 0.170 | 0.146 | 0.198 | 0.120 | 0.130 | 0.115 | 0.127 | 0.128 | 0.130 | 0.120 | **0.115** | 0.270 | 1.430 | 0.552 |
| | | medium | 0.160 | 0.157 | 0.134 | 0.155 | 0.120 | 0.128 | 0.127 | 0.121 | 0.118 | 0.127 | 0.118 | **0.114** | 0.255 | 1.210 | 0.479 |
| | | short | 0.155 | **0.125** | 0.133 | 0.140 | 0.132 | 0.133 | 0.128 | 0.139 | 0.148 | 0.128 | 0.173 | 0.139 | 0.281 | 0.549 | 0.281 |
| restaurant | D | short | 0.263 | 0.264 | 0.264 | 0.276 | 0.260 | 0.286 | **0.255** | 0.261 | 0.297 | 0.270 | 0.262 | 0.284 | 0.362 | 0.329 | 0.907 |
| saugeen | D | short | 0.373 | 0.338 | 0.354 | 0.339 | **0.328** | 0.379 | 0.358 | 0.408 | 0.353 | 0.572 | 0.408 | 0.419 | 0.564 | 0.669 | 0.754 |
| | M | short | **0.276** | 0.296 | 0.293 | 0.288 | 0.291 | 0.332 | 0.297 | 0.342 | 0.299 | 0.689 | 0.372 | 0.340 | 0.326 | 0.373 | 0.445 |
| | W | short | 0.395 | 0.363 | 0.372 | 0.364 | 0.419 | 0.406 | **0.350** | 0.601 | 0.390 | 0.397 | 0.384 | 0.491 | 0.549 | 0.734 | 0.855 |
| solar | 10T | long | 0.331 | 0.443 | 0.497 | 0.534 | 0.408 | 0.365 | **0.321** | 0.498 | 0.352 | 0.549 | 0.339 | 0.379 | 0.786 | 6.640 | 0.786 |
| | | medium | **0.326** | 0.436 | 0.453 | 0.495 | 0.412 | 0.373 | 0.334 | 0.516 | 0.353 | 0.485 | 0.356 | 0.362 | 0.771 | 5.670 | 0.771 |
| | | short | 0.458 | 0.511 | 0.498 | 0.488 | **0.350** | 0.444 | 0.541 | 0.804 | 0.541 | 0.933 | 1.370 | 0.618 | 0.860 | 2.360 | 0.860 |
| | D | short | **0.269** | 0.287 | 0.286 | 0.282 | 0.303 | 0.324 | 0.273 | 0.278 | 0.290 | 0.682 | 0.287 | 0.277 | 0.282 | 0.286 | 0.757 |
| | H | long | 0.351 | 0.405 | 0.373 | 0.354 | 0.332 | 0.293 | **0.287** | 0.493 | 0.331 | 0.381 | 0.353 | 0.401 | 0.607 | 7.320 | 1.470 |
| | | medium | 0.313 | 0.368 | 0.356 | 0.342 | 0.333 | 0.309 | **0.285** | 0.376 | 0.331 | 0.352 | 0.344 | 0.330 | 0.557 | 6.130 | 1.270 |
| | | short | 0.336 | 0.298 | 0.303 | 0.313 | 0.342 | 0.329 | **0.273** | 0.406 | 0.328 | 0.389 | 0.340 | 0.367 | 0.628 | 2.330 | 0.628 |
| | W | short | 0.120 | 0.133 | 0.136 | 0.132 | 0.163 | 0.148 | 0.144 | 0.171 | 0.186 | 0.242 | 0.162 | **0.114** | 0.152 | 0.155 | 0.236 |
| sz_taxi | 15T | long | 0.242 | 0.248 | 0.245 | 0.240 | 0.217 | 0.221 | **0.198** | 0.227 | 0.202 | 0.286 | 0.281 | 0.241 | 0.398 | 0.629 | 0.554 |
| | | medium | 0.230 | 0.244 | 0.246 | 0.240 | 0.214 | 0.228 | **0.202** | 0.229 | 0.205 | 0.210 | 0.220 | 0.206 | 0.351 | 0.529 | 0.454 |
| | | short | 0.209 | 0.202 | 0.203 | 0.203 | 0.201 | 0.223 | 0.200 | **0.199** | 0.203 | 0.219 | 0.207 | 0.222 | 0.309 | 0.288 | 0.309 |
| | H | short | 0.140 | 0.136 | 0.137 | 0.137 | 0.136 | 0.154 | 0.135 | **0.135** | 0.137 | 0.139 | 0.144 | 0.144 | 0.170 | 0.232 | 0.229 |
| temperature_rain | D | short | 0.569 | **0.538** | 0.544 | 0.548 | 0.561 | 0.620 | 0.547 | 0.586 | 0.560 | 0.682 | 0.644 | 0.592 | 0.694 | 0.761 | 1.630 |
| us_births | D | short | **0.016** | 0.026 | 0.028 | 0.037 | 0.020 | 0.022 | 0.023 | 0.019 | 0.026 | 0.028 | 0.025 | 0.016 | 0.074 | 0.075 | 0.144 |
| | M | short | 0.015 | 0.019 | 0.016 | 0.017 | 0.017 | 0.028 | 0.012 | 0.011 | 0.013 | 0.016 | 0.017 | 0.021 | **0.010** | 0.019 | 0.017 |
| | W | short | **0.011** | 0.013 | 0.013 | 0.014 | 0.011 | 0.017 | 0.012 | 0.013 | 0.014 | 0.017 | 0.015 | 0.019 | 0.018 | 0.018 | 0.022 |

### A.6 Covariate-Informed Forecasting Evaluation on `fev-bench`

We evaluate on the subset of `fev-bench` tasks that include covariates in both the conditioning and prediction windows (commonly referred to in the time-series literature as past and known future covariates). The tasks used are listed in Table 4. Detailed metadata are provided in the original benchmark paper (Shchur et al., 2025). As these details are already standardized and publicly available, we refer readers to the benchmark documentation rather than replicate the full tables here.

| # | `fev-bench` Tasks |
|---|---|
| 1 | proenfo_gfc12 |
| 2 | proenfo_gfc14 |
| 3 | proenfo_gfc17 |
| 4 | rohlik_orders_1D |
| 5 | rohlik_sales_1W |
| 6 | rohlik_orders_1W |
| 7 | entsoe_15T |
| 8 | entsoe_30T |
| 9 | entsoe_1H |
| 10 | epf_be |
| 11 | epf_de |
| 12 | epf_fr |
| 13 | epf_np |
| 14 | epf_pjm |
| 15 | rossmann_1D |
| 16 | rossmann_1W |
| 17 | hermes |
| 18 | walmart |
| 19 | m5_1W |
| 20 | m5_1M |
| 21 | favorita_stores_1D |
| 22 | favorita_stores_1W |
| 23 | favorita_stores_1M |
| 24 | favorita_transactions_1D |
| 25 | solar_with_weather_15T |
| 26 | solar_with_weather_1H |
| 27 | uci_air_quality_1H |
| 28 | uci_air_quality_1D |

Table 4: Subset of `fev-bench` tasks used for covariate-informed forecasting. Detailed task characteristics are available in the original benchmark paper (Shchur et al., 2025).

## A.7    Additional Results for Harmonic Reconstruction Experiments

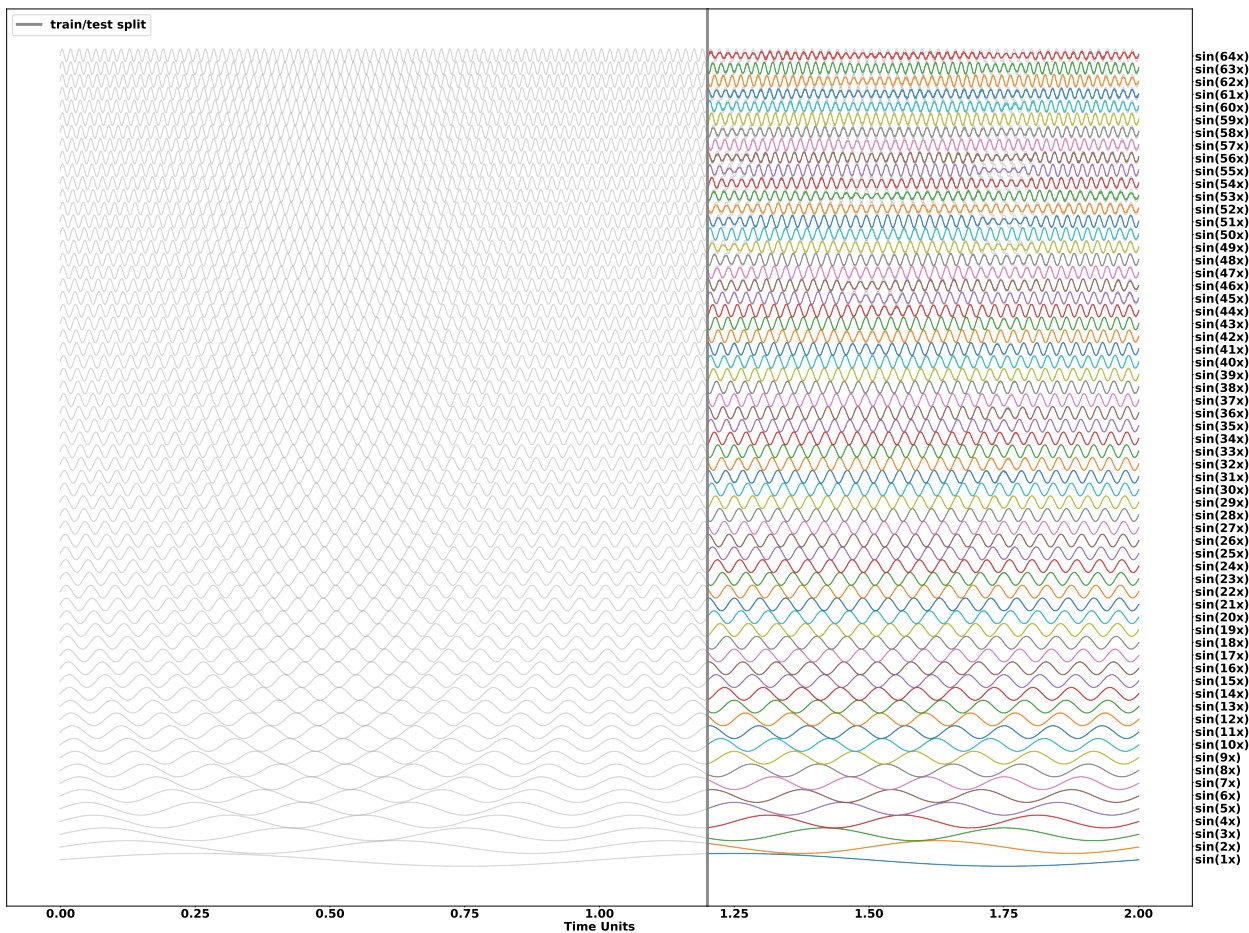

Figure 14: Reconstruction of $\sin(nt)$ under the injective temporal embedding $\{\sin(t), \cos(t)\}$. TabPFN-v2 is evaluated on $\sin(nt)$ for $n = 1, \ldots, 64$. The model accurately reconstructs low and mid-range frequencies, but accuracy gradually degrades at higher n, consistent with the finite phase resolution of the sinusoidal embedding given the sampling density.

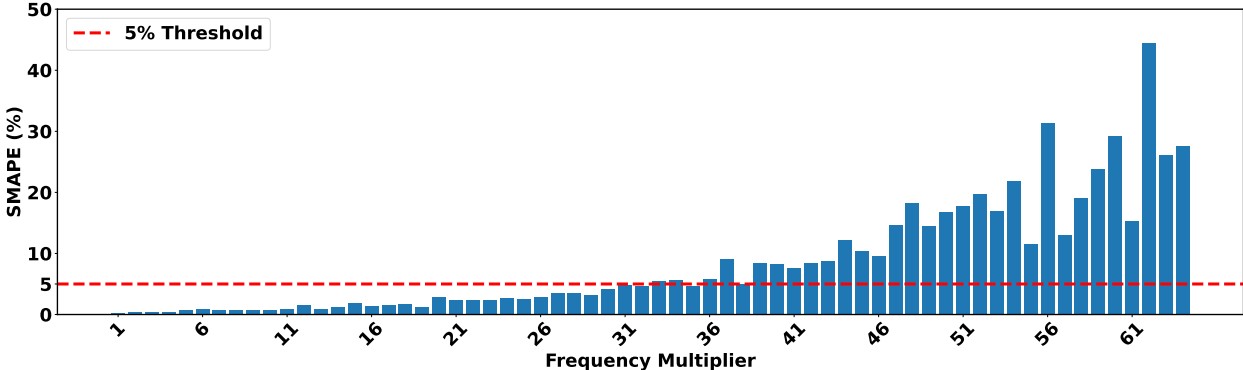

Figure 15: Symmetric mean absolute percentage error (sMAPE) for predicting $\sin(nt)$ using $\{\sin(t), \cos(t)\}$ as input features. Errors remain low for moderate frequencies but increase as n grows, reflecting the reduced separability of high-frequency phases in the sinusoidal coordinate system.

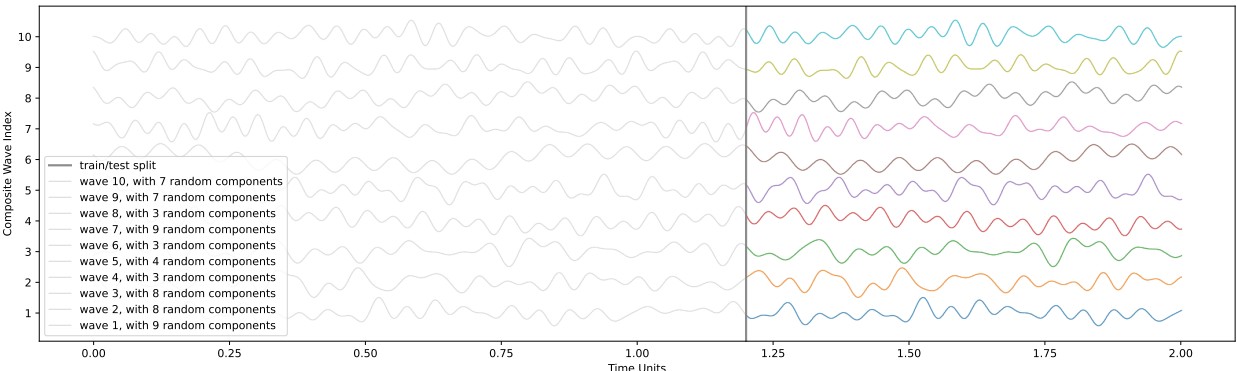

Figure 16: Reconstruction of composite sinusoidal signals under the injective temporal embedding $\{\sin(t), \cos(t)\}$. Each composite signal is the sum of 3–10 sinusoids with random frequencies, amplitudes, and phases. TabPFN-v2 accurately recovers these multi-frequency patterns, consistent with interpolation in a phase-preserving temporal embedding.

### A.7.1 Phase Ambiguity and the Requirement of Injective Mapping

**The Non-Injectivity of** $\sin(t)$**.** When time $t$ is mapped via the feature $\phi(t) = \sin(t)$, the representation is non-injective. Within a single period $T = 2\pi$, every feature value $y \in (-1, 1)$ corresponds to two distinct phases–specifically $t$ and $\pi - t$–creating a *phase ambiguity*. To a retrieval-based model, these points are indistinguishable, resulting in a coordinate collision where the rising and falling edges of a cycle share the same address in feature space.

**Symmetry and Harmonic Recovery.** The capacity to recover higher-order harmonics from this ambiguous base depends on the symmetry of the target signal $y_t = \sin(nt)$ relative to these collision points:

- Odd Harmonics (Consistent Mapping): For odd $n$, the signal is symmetric across the ambiguity: $y(t) = y(\pi - t)$. Consequently, both time points that share a single $\sin(t)$ feature value also share an identical target value. This ensures that the mapping from the feature space to the target remains consistent, allowing the model to recover the signal despite the underlying phase ambiguity.

- Even Harmonics (Inconsistent Mapping): For even $n$, however, the signal is anti-symmetric: $y(t) = -y(\pi - t)$. This presents the model with a mathematical contradiction–two distinct target values mapped to a single feature coordinate–hence the failure to recover even-ordered harmonics.

**Quadrature Resolution via** $(\sin(t), \cos(t))$**.** Providing the quadrature pair $\Phi(t) = (\sin(t), \cos(t))$ embeds time onto the unit circle $S^1 \subset \mathbb{R}^2$. This mapping is injective: it assigns a unique coordinate to every phase within the period. Consequently, the model can uniquely associate any target value—regardless of harmonic order—with its corresponding phase.

### A.7.2 Discussion of Functional Synthesis vs. Phase-Space Retrieval

In this section, we investigate two competing hypotheses for how the model reconstructs higher-order harmonics ($y_t = \sin(nt)$) from base periodic features ($\sin(t), \cos(t)$).

1. *The Functional Approximation Hypothesis:* The model uses its internal MLP layers to compute trigonometric identities or polynomial expansions (similar to a Chebyshev expansion) to "synthesize" higher frequencies from the base inputs. An example would be, approximating

$$\sin(4x) = 4\sin(x)\cos^3(x) - 4\sin^3(x)\cos(x).$$

2. *The Phase-Space Retrieval Hypothesis:* The model treats ($\sin(t), \cos(t)$) as a unique coordinate (an injective "key") and uses the attention mechanism to retrieve and interpolate values from the observed context window.

**The Half-Cycle Experiment.** To probe these competing explanations, we test the model's ability to extrapolate beyond observed feature ranges. If the model were performing global functional synthesis, it should theoretically be capable of generalizing the relationship $y_t = f(\sin(t), \cos(t))$ to any input value, as the underlying trigonometric identities are globally valid regardless of the observed range.

Similar to Section 5.2, we task the model with reconstructing higher-order harmonics ($y_t = \sin(nt)$) from base periodic features ($\sin(t), \cos(t)$). However, we restrict the support set to $t \in [0, \pi]$. In this setting, the model never observes a full cycle of the ($\sin(t), \cos(t)$) coordinate system in context, which is a prerequisite for phase-space retrieval.

As shown in Figure 17, the model fails to reconstruct the signals in the unobserved range. If the model were performing polynomials expansions, it could theoretically infer the relationship from the mathematically sufficient, albeit partial, view of the mapping $t \mapsto (\sin(t), \cos(t))$.

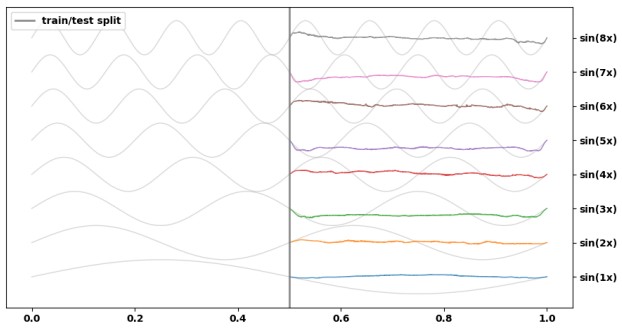

Figure 17: The Half-Cycle Experiment. We evaluate the model's ability to reconstruct harmonics $\sin(nt)$ when the support set features ($\sin t, \cos t$) are restricted to a partial cycle ($t \in [0, \pi]$). The failure of recovering the harmonics in the unseen phase range ($\pi, 2\pi$] suggests that the model does not generalize via global trigonometric identities.

While these results do not strictly rule out more complex forms of functional approximation, they provide empirical evidence that TabPFN-v2 is less likely to be performing global functional synthesis in this context. Instead, the failure to extrapolate beyond the observed coordinates point towards the model's proficiency being tied to its behavior as a context-dependent retrieval engine.

**Conclusions.** The evidence suggests that TabPFN-TS handles seasonal signals by operating as a **meta-learned retrieval engine**. By providing ($\sin(t), \cos(t)$), we supply a coordinate system that minimizes "collisions" (non-injectivity), This enables TabPFN-v2 to uniquely and accurately retrieve relevant seasonal patterns from the series history, provided those coordinates have been previously observed in the support set.

### A.8 Synthetic Data Generation for Time Series with Covariates

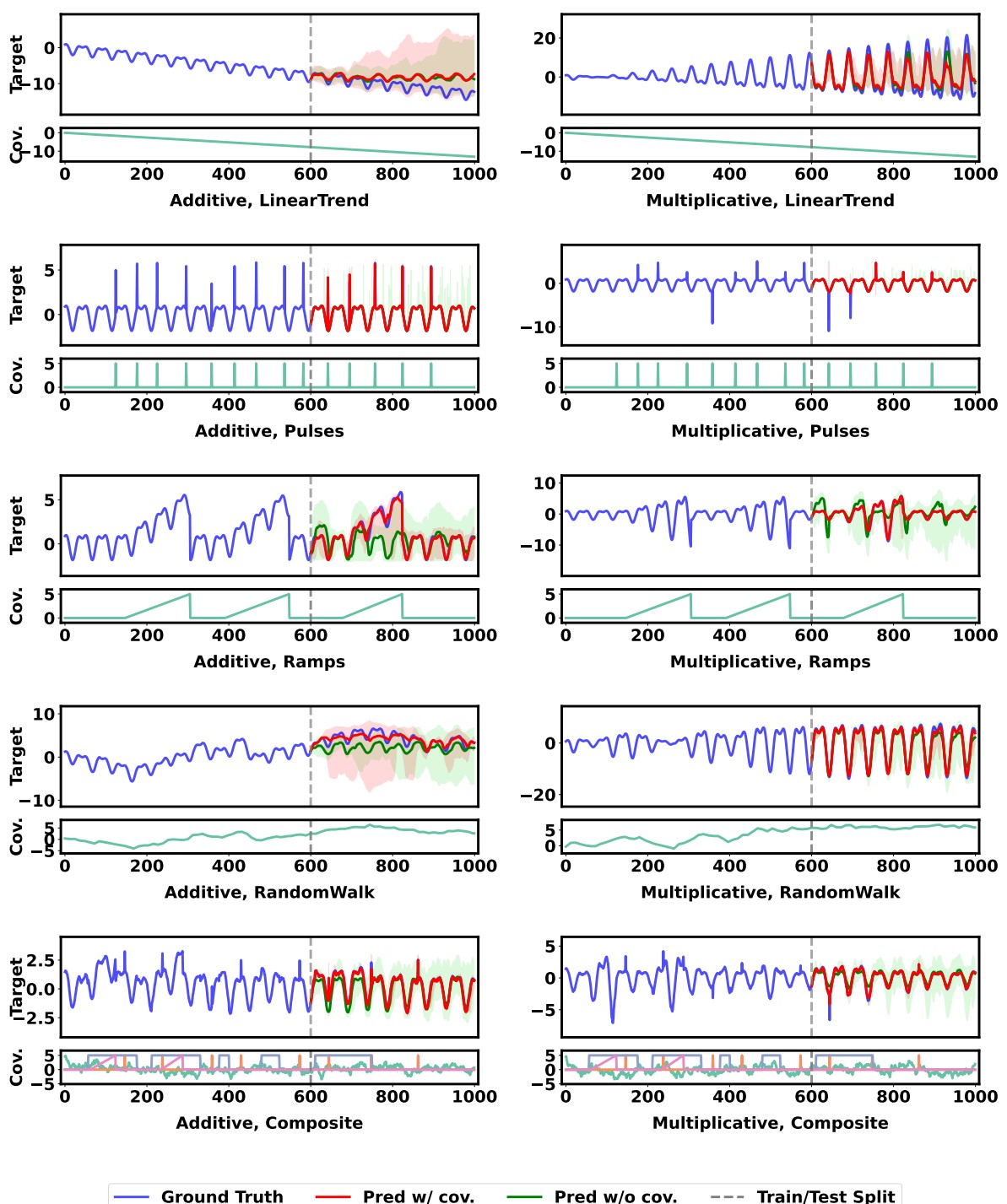

Figure 18: Forecasting visualization on synthetic covariate-augmented time series. Each subplot corresponds to one synthetic class from $\mathcal{C}$ = additive, multiplicative $\times$ LinearTrend, Pulses, Ramps, RandomWalk, Composite. Blue lines denote ground truth, red lines predictions with covariates, and green lines predictions without covariates. Shaded regions indicate predictive uncertainty, and the vertical dashed line marks the train/test split.

## A.9 Ablation: Context Length vs. Accuracy

In this ablation, we investigate how the amount of available context affects the performance of TabPFN-TS. We experiment with four context lengths: 1024, 2048, 4096, and 10,000. The maximum length of 10,000 is chosen to match the largest dataset size used during the pretraining of TabPFN-v2.

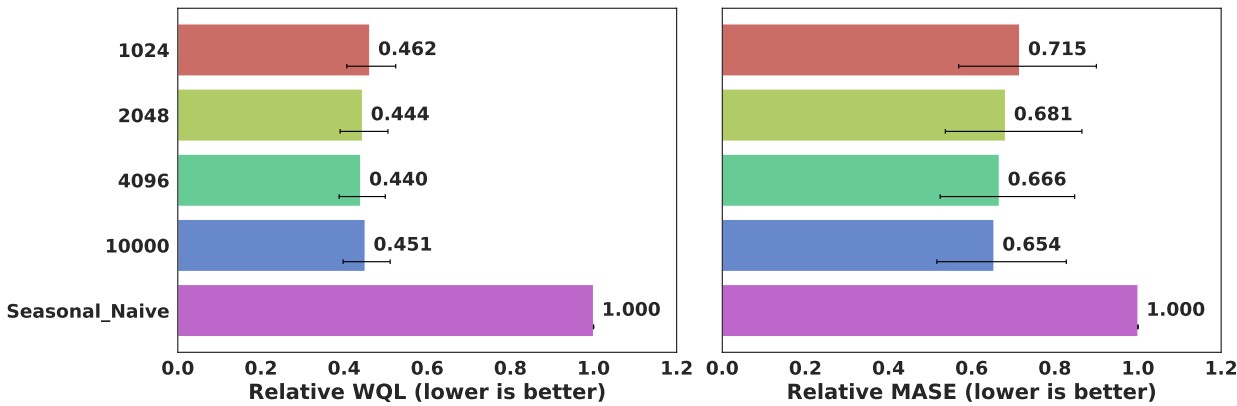

Figure 19: Effect of context length on forecasting performance of TabPFN-TS.

As shown in Figure 19, increasing the context length leads to improved performance overall, though the gains diminish beyond 4096 points. While MASE continues to improve with longer context, WQL shows a slight increase at the longest length. These results suggest that moderate-length contexts are often sufficient, but the impact of longer contexts may vary depending on the forecasting objective.

## A.10 Probablistic Calibration Analysis

To complement Weighted Quantile Loss (WQL, see Section 4), we assess calibration directly, since WQL conflates calibration with sharpness. We report average coverage error[4] (Alexandrov et al., 2020), following the calibration framework of Gneiting et al. (2007):

$$\text{ACE} = \frac{1}{|Q|} \sum_{q \in Q} |\text{Cov}(q) - q|, \qquad \text{Cov}(q) = \frac{1}{N} \sum_{i=1}^{N} \mathbb{1}\!\!\!\!/\!\left[y_i \leq \hat{Q}_i(q)\right],$$

where $Q = \{0.1, \ldots, 0.9\}$, $\hat{Q}_i(q)$ is the predicted $q$-quantile for test point $i$, and $N$ ranges over all series and timesteps in a dataset. ACE is averaged across 81 smallest tasks (out of 97) in GIFT-Eval.

Figure 20 reports ACE for TabPFN-TS and the time-series foundation model baselines for which quantile forecasts were available.

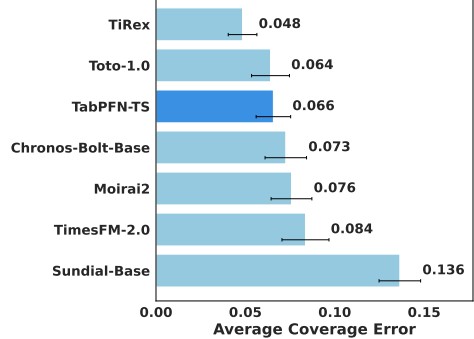

Figure 20: Average coverage error (ACE, lower is better) across 81 tasks in GIFT-Eval. Error bars show 95% confidence intervals.

TiRex attains the lowest ACE (0.048), with confidence intervals separating it from the remaining models. TabPFN-TS (0.066) falls within a cluster of comparable values alongside Toto-1.0 (0.064), Chronos-Bolt-Base (0.073), and Moirai2 (0.076), whose confidence intervals overlap substantially with one another, while Sundial-Base (0.136) is a clear outlier on the high end. These results indicate that TabPFN-TS's predictive uncertainty quality is comparable to most existing time-series foundation models, though not as good as TiRex.

---

[4]https://ts.gluon.ai/dev/api/gluonts/gluonts.ev.metrics.html#gluonts.ev.metrics.MAECoverage

### A.11   Inference Optimization Strategies

The inference cost of TabPFN-TS is higher than competing TSFMs, as discussed in Section 6.2. Here we outline potential strategies that could reduce this overhead in future work.

**Patch-based sequence compression.** The most impactful optimization would be to introduce patch-based tokenization of the input sequence, analogous to Chronos-Bolt and PatchTST. Rather than passing each time step as an individual row to TabPFN-v2, consecutive time steps could be aggregated into patches, reducing the effective sequence length by a factor of the patch size $p$. For a context of length $L$ and patch size $p$, this reduces the attention cost from $\mathcal{O}(L^2)$ to $\mathcal{O}((L/p)^2)$ — a reduction of $p^2$. For $p = 16$ and $L = 4096$, this yields a $256\times$ reduction in attention cost. Beyond reducing inference cost, patching would also enable series batching (see below) by making memory consumption per series tractable.

However, introducing patching would require either retraining a patch encoder on time series data or finding a patch aggregation strategy compatible with the existing TabPFN-v2 architecture (e.g. mean pooling of temporal features within each patch).

**Series batching.** Currently, TabPFN-TS executes a separate forward pass for each individual time series. A natural optimization is to batch multiple series into a single forward pass, amortizing the fixed overhead across many series and improving throughput without any architectural changes. However, naively batching full-length series would significantly increase memory consumption due to the quadratic memory cost of attention. This makes batching most practical in conjunction with patch-based compression, which would bring per-series memory to a tractable level.

**Sparse and linear attention.** The quadratic attention cost of TabPFN-v2 could be addressed through sparse or linear attention mechanisms. However, since TabPFN-v2's attention operates over tabular rows rather than sequence tokens, standard sparse attention patterns (e.g. local windows, strided attention) may not directly apply. Developing attention approximations compatible with the tabular in-context learning objective of TabPFN-v2 remains an open research direction.

**Immediate workaround: data-parallel inference.** We note that all of the above optimizations require architectural changes and/or retraining, and therefore do not provide an immediate speed-up. In practice, the most straightforward way to reduce wall-clock inference time is data-parallel evaluation across multiple GPUs, where each GPU processes an independent subset of series. This is the approach used in our experiments, and scales linearly with the number of available GPUs. While this does not reduce the per-series computational cost, it provides a practical workaround for large-scale evaluation.

