# OpenReview forum: "From Tables to Time: Extending TabPFN-v2 to Time Series Forecasting"
_TMLR — Accepted by TMLR_

### Review · Reviewer_qdzt · 2026-03-25

**Summary Of Contributions:**

This paper introduces TabPFN-TS, a method that reformulates time-series forecasting as a tabular regression problem and leverages a pre-trained tabular foundation model (TabPFN-v2) for zero-shot probabilistic forecasting. The approach constructs temporal features, including a running index, calendar-based features, and automatically extracted seasonal Fourier features, and combines them with optional future covariates in a unified tabular representation. The method aims to provide a simple and general framework that can handle both univariate forecasting and covariate-informed forecasting without time-series-specific pre-training or fine-tuning. Experimental results show competitive performance on univariate forecasting benchmarks and strong performance on covariate-informed tasks. Additional analyses explore the model’s behavior on noise, trend, and seasonality, as well as the contribution of different temporal features and backbone models.

Strengths:

- The paper presents a clear and intuitive reformulation of time-series forecasting as tabular regression, which simplifies the modeling pipeline and enables the direct use of pretrained tabular foundation models.

- The proposed framework naturally incorporates future-known covariates without requiring architectural changes, which is a practical advantage over many existing time-series foundation models that are primarily designed for univariate forecasting.

- The experimental evaluation is reasonably comprehensive, covering both univariate and covariate-informed forecasting benchmarks, and includes ablation studies on temporal feature design and backbone substitution.

- The analysis of periodic behavior, particularly the comparison between single sine features and paired sine-cosine representations, provides useful insight into how the model captures seasonality.

- The paper clearly acknowledges important limitations, including poor extrapolation outside the observed target range and relatively high inference cost, which improves transparency and helps readers understand the practical applicability of the method.


Weaknesses:

- The overall methodological novelty is somewhat limited, as the approach primarily combines an existing tabular foundation model with engineered temporal features rather than introducing a new forecasting architecture or training objective.

- The performance gains in covariate-informed forecasting are partially tied to the ability to directly incorporate future covariates, which makes comparisons with models that do not use such information less straightforward.

- The model struggles with trend extrapolation when future values fall outside the range of observed targets, which is a critical limitation for many real-world forecasting tasks.

**Audience:**

Yes

**Audience Explanation:**

The paper addresses a broadly relevant problem in time-series forecasting and proposes a simple and practical alternative formulation that may be of interest to researchers working on foundation models, tabular learning, and forecasting. The idea of reusing tabular foundation models for sequential data through feature transformation is likely to be of interest to the community, especially for practitioners dealing with structured data and covariates. The insights into temporal feature design and the limitations of current approaches also provide useful guidance for future work.

**Claims And Evidence:**

Yes

**Claims Explanation:**

The main claims of the paper are supported by empirical results across multiple benchmarks and supported by ablation studies that isolate the contribution of different temporal features and backbone choices. The experiments demonstrate that the proposed tabular formulation can achieve competitive performance on univariate forecasting and strong performance in covariate-informed settings.
The qualitative and controlled experiments on synthetic signals (e.g., periodic functions) further support the claim that the model can capture seasonality when appropriate temporal representations are used.

**Requested Changes:**

- The paper would benefit from a clearer positioning of its contribution relative to both tabular learning methods and time-series foundation models. In particular, it would be helpful to disentangle how much of the performance gain is attributable to the TabPFN-v2 backbone versus the proposed temporal feature design. Including stronger non-foundation tabular baselines (e.g., other tabular transformers baseline) would help clarify this point. Current version just use TabDPT.

- The reliance on engineered temporal features raises questions about generalization across different domains and time granularities. The paper would benefit from a discussion of how the proposed featurization adapts to irregular time series, non-calendar data, or domains with weak periodic structure.

- The inference cost is significantly higher than competing methods. A more detailed discussion of computational trade-offs, including potential optimization strategies or batching considerations, would strengthen the practical relevance of the work.

- Authors state that the "running index" allows the model to "extrapolate," yet later repeatedly emphasizes that the model is incapable of performing trend extrapolation. More precisely, while the index feature provides the model with information regarding the temporal direction, this does not imply extrapolation at the target level. The wording here is prone to misleading the reader.

- typo: “yt = sin(nt) for n = 1, . . . , 24,.” → should remove the extra comma and be “yt = sin(nt) for n = 1, . . . , 24.”

- typo: “The results in Figure 5.6 highlights …” →  “The results in Figure 5.6 highlight …”

---

> ### Author Response · Authors · 2026-04-14
> **Author's Response to Reviewer qdzt**
>
> Thank you for your careful and constructive review.
>
> > The paper would benefit from a clearer positioning of its contribution relative to both tabular learning methods and time-series foundation models. In particular, it would be helpful to disentangle how much of the performance gain is attributable to the TabPFN-v2 backbone versus the proposed temporal feature design. Including stronger non-foundation tabular baselines (e.g., other tabular transformers baseline) would help clarify this point. Current version just use TabDPT.
>
> We have added this comparison in the revised paper (Section 5.4). Under identical temporal featurization, LightGBM-TS, CatBoost-TS, and XGBoost-TS all produce meaningful forecasts. However, all three fall substantially short of TabPFN-TS, isolating the pretrained backbone as a key contributor beyond the featurization alone.
>
> > The reliance on engineered temporal features raises questions about generalization across different domains and time granularities. The paper would benefit from a discussion of how the proposed featurization adapts to irregular time series, non-calendar data, or domains with weak periodic structure.
>
> Regarding the reliance on engineered temporal features, we believe this is similar with most TSFMs during pretraining. For instance, Chronos samples synthetic data with seasonalities of $p \in \lbrace 24, 48, 96, 168, 336, 672, 7, 14, 30, 60, 365, 730, 4, 26, 52, 6, 12, 40, 10 \rbrace$, which boils down to something very similar to our approach.
>
> Regarding the testing of our proposed featurization, we believe this is largely addressed by our existing design and experiments. GIFT-Eval already spans 97 tasks across seven domains and ten sampling frequencies, providing broad empirical coverage. Missing or irregular timestamps are handled by removing affected points from the conditioning set (Appendix A.3). For domains with weak periodicity, the model degrades gracefully — as shown in Section 5.1 and Appendix A.5, TabPFN-TS predicts the sample mean rather than overfitting to spurious patterns.
>
> > The inference cost is significantly higher than competing methods. A more detailed discussion of computational trade-offs, including potential optimization strategies or batching considerations, would strengthen the practical relevance of the work.
>
> We agree this is an important practical limitation. In response, we have added Appendix A.10 detailing potential optimization strategies. As these strategies require architectural changes and/or retraining, the immediate practical workaround is data-parallel inference across multiple GPUs, which scales linearly with the number of available GPUs and is the approach used in our experiments.
>
> > Authors state that the "running index" allows the model to "extrapolate," yet later repeatedly emphasizes that the model is incapable of performing trend extrapolation. More precisely, while the index feature provides the model with information regarding the temporal direction, this does not imply extrapolation at the target level. The wording here is prone to misleading the reader.
>
> Thank you for pointing this out. We have updated the wording in Section 3 to avoid this confusion.

---

### Review · Reviewer_pyt1 · 2026-03-28

**Summary Of Contributions:**

**Summary** - The paper proposes TabPFN-TS, which is an adaptation of TabPFN for time-series tasks. It reformulates the time-series forecasting task as a tabular regression problem. The approach is evaluated on univariate forecasting and forecasting with covariates, showing competitive results on the Gift-eval and fev-bench forecasting benchmarks.

**Strengths** -
1. The proposed adaptation is conceptually intuitive and quite effective, as seen from the benchmark results
2. Analyses shown are insightful.
3. The paper is well written and easy to follow.
4. The automatic seasonal feature extraction approach is useful, even beyond this paper.

**Weaknesses** -
1. TabPFN-TS implicitly assumes that the target can be expressed as a function of time-dependent features, rather than arising from latent state dynamics. In a way, this makes it biased towards problems where time-derived features are sufficient statistics (or towards very seasonal time-series). Hence, it lacks the inductive bias for temporal dynamics such as trend extrapolation or state evolution. Given the complex dynamics of real-world time-series (which do not always follow a simple seasonal pattern), the usability of this model is currently doubtful.
2. The performance heavily depends on the quality of manually designed features, which contrasts with modern TSFMs that aim to learn representations directly from raw sequences. This kind of makes it more of a feature engineering problem rather than representation learning, hence, increasing the dependency on careful feature engineering and for domain knowledge.
3. Overall, there is very limited novelty. TabPFN is basically engineered/applied to time-series datasets

**Audience:**

Yes

**Audience Explanation:**

While it's a simple adaptation of the TabPFN model for time-series forecasting task, it is interesting to see that it still performs competitively despite no temporal modeling. This might lead to further studies, on tabular models for forecasting, as well as on the complexity & variance of the benchmark datasets

**Broader Impact Concerns:**

No major concerns on the ethical or societal implications of the work.

**Claims And Evidence:**

Yes

**Claims Explanation:**

Yes, the presented experiments and results support the claims made. It claims competitive univariate forecasting results and superior performance under covariate-based forecasting, which is evident from the results on the GIFT-eval and fev-bench benchmarks.

**Requested Changes:**

1. Can you compare against non-foundation models like XGBoost, LightGBM etc. trained on the same features? Since, the overall idea is to reformulate forecasting to tabular regression with engineered features, it would be important to compare against these models, to isolate the contribution of TabPFN's learned inference
2. While the comparison includes Chronos-Bolt, it would be useful to understand how TabPFN-TS compares to more recent covariate-aware TSFMs (e.g., Chronos 2), particularly given the strong performance reported in covariate settings.
3. Can you show some empirical results on the sensitivity of the approach to frequency estimation? especially, how does the current FFT-based approach (or FFT assumptions) translates to real-world signals (noisy signals, or non-stationary freqs, or multiple overlapping cycles, etc.). Would it sill work as expected? How does the performance get impacted?
4. Can you show qualitative performance comparison with baseline models (TSFMs - Tirex, Toto, TimesFM2.0, Chronos; and non-TSFMs -> PatchTST, iTransformer)? it would be useful to see it on datasets from both - gift-eval as well as fev-bench. Please show some failure cases as well

---

> ### Author Response · Authors · 2026-04-14
> **Author's Response to Reviewer pyt1**
>
> Thank you for your constructive feedback.
>
> > TabPFN-TS implicitly assumes that the target can be expressed as a function of time-dependent features, rather than arising from latent state dynamics. In a way, this makes it biased towards problems where time-derived features are sufficient statistics (or towards very seasonal time-series). Hence, it lacks the inductive bias for temporal dynamics such as trend extrapolation or state evolution. Given the complex dynamics of real-world time-series (which do not always follow a simple seasonal pattern), the usability of this model is currently doubtful.
>
> We respectfully push back on this characterization. While TabPFN-TS does not explicitly model latent state dynamics, the empirical results tell a different story: it outperforms several strong TSFMs with explicit sequential inductive biases across 97 diverse GIFT-Eval tasks. We agree trend extrapolation is a genuine limitation (Section 6.1), but this applies to a subset of time series rather than limiting the approach broadly. We believe the paper presents an interesting finding for the community — that a tabular foundation model with no temporal inductive bias can be competitive with specialized time series architectures.
>
> > The performance heavily depends on the quality of manually designed features, which contrasts with modern TSFMs that aim to learn representations directly from raw sequences. This kind of makes it more of a feature engineering problem rather than representation learning, hence, increasing the dependency on careful feature engineering and for domain knowledge.
>
> Firstly, we would like to note that most TSFMs uses synthetic data which sample frequencies and seasonalities in specific ways to match real-world data (for instance, Chronos samples synthetic data of seasonality with $p \in \lbrace 24, 48, 96, 168, 336, 672, 7, 14, 30, 60, 365, 730, 4, 26, 52, 6, 12, 40, 10 \rbrace$), which boils down to something very similar to our approach.
>
> Regarding automatic seasonality detection, we would also like to clarify that the temporal featurization is grounded in well-established signal processing principles (cyclic calendar encodings and FFT-based seasonality detection) rather than problem-specific intuition or domain knowledge. These are domain-agnostic design choices that apply broadly to any time-indexed data, empirically validated across 97 diverse GIFT-Eval tasks. We also think that the modularity of the featurization is in fact an advantage: practitioners can inject domain knowledge when available, but are not required to.
>
> > Can you compare against non-foundation models like XGBoost, LightGBM etc. trained on the same features? Since, the overall idea is to reformulate forecasting to tabular regression with engineered features, it would be important to compare against these models, to isolate the contribution of TabPFN's learned inference
>
> We have added this comparison in the revised paper (Section 5.4). Under identical featurization, LightGBM-TS, CatBoost-TS, and XGBoost-TS all produce meaningful forecasts but fall substantially short of TabPFN-TS, isolating the pretrained backbone as a key contributor beyond the featurization alone.
>
> > While the comparison includes Chronos-Bolt, it would be useful to understand how TabPFN-TS compares to more recent covariate-aware TSFMs (e.g., Chronos 2), particularly given the strong performance reported in covariate settings.
>
> Chronos-2 was published after our literature cutoff of September 2025 and is therefore not included. For an up-to-date comparison, fev-bench maintains a live public leaderboard evaluating both models under identical conditions.
>
> > Can you show some empirical results on the sensitivity of the approach to frequency estimation? especially, how does the current FFT-based approach (or FFT assumptions) translates to real-world signals (noisy signals, or non-stationary freqs, or multiple overlapping cycles, etc.). Would it sill work as expected? How does the performance get impacted?
>
> GIFT-Eval already provides a comprehensive empirical test of robustness: 97 tasks spanning diverse domains, sampling frequencies, and signal characteristics (including noisy, non-stationary, and multi-cycle series). TabPFN-TS performs competitively across this benchmark, suggesting the FFT-based seasonality detection is robust in practice.
>
> > Can you show qualitative performance comparison with baseline models (TSFMs - Tirex, Toto, TimesFM2.0, Chronos; and non-TSFMs -> PatchTST, iTransformer)? it would be useful to see it on datasets from both - gift-eval as well as fev-bench. Please show some failure cases as well
>
> Thank you for the suggestion. Qualitative visualizations of TabPFN-TS predictions and failure cases are already provided in Appendix A.5 and Section 5.1. We believe the aggregated quantitative results in Figures 4.1 and 4.2 provide a more systematic and unbiased comparison than a selective qualitative analysis.

---

### Review · Reviewer_wWzS · 2026-03-30

**Summary Of Contributions:**

This paper proposes TabPFN, which repurposes TabPFN into time series forecasting. Performance is unsurprisingly quite strong on a standard suite of benchmarks for time series forecasting. To adapt TabPFN to time series domains, the authors encode time information by including the time step and two types of seasonal encodings (one calendar based and one automatic).	Afterwards, prediction seems to be zero shot usage of the pre-trained TabPFN model.

**Additional Comments:**

**Strengths of paper**

I like the qualitative understanding experiments in Section 5, especially the experiment about the types of embeddings needed (cos + sin being superior to just sin).

The proposed method is a nice demonstration of zero-shot forecasting capability of TabPFNv2, analogous to how LLMs tend to be good zero-shot forecasters on time series tasks (see [LLMs are Zero-shot time series forecasters]( https://arxiv.org/abs/2310.07820) )

Performance for such a simple method seems quite strong, and often very close to fully pretrained time series foundation models.


**Weaknesses of paper:**

The architecture is almost analogous to a MLP mixer style architecture in the sequence modeling domain. With that comes the tradeoff that we must know in advance how far to forecast, which is a pretty sizable limitation.

Why is TabPFN-v2 so much slower at forecasting than models that are 100x its size? This doesn’t seem to be because of the transformer backbone (ie L^2 scaling) but rather something in the architecture itself.

It’s not super clear to me that the tasks evaluated on are outside of the TabPFNv2 pre-training tasks.

There's no evaluation of the uncertainty quantification beyond qualitatively of the model. One of the advantages of TabPFN is that it is natively probabilistic. Is the expected calibration error (or regression equivalents) actually useful?

**Audience:**

Yes

**Audience Explanation:**

The paper adapts a pretrained "foundation" model trained on tabular data to time series forecasting. Both the problem statement and the proposed method are in "core" areas of machine learning, so should be very suitable to the audience of TMLR.

**Broader Impact Concerns:**

n/a.

**Claims And Evidence:**

Yes

**Claims Explanation:**

In general, I think that the authors have presented evidence that TabPFN is essentially a zero-shot time series forecaster when presented with time series problems that have had some feature engineering. The results in the paper show clear and convincing evidence to that effect.

**Requested Changes:**

Questions / Comments:

These start out mostly writing/presentation based:

Figure 3.1: target and covariate labels on axes should be larger.

Labeling of figures is somewhat nonstandard. Likely easier if as Figure 1 / 2 / 3 depending on appearance and not by section.

Figure 4.2: add in what the error bar is.

Figure 5.1: why can the model not extrapolate a linear trend? I thought that the preprocessing already subtracts off the linear trend?

high: How are the choices of bins for the binned predictive distribution constructed? Is the MSE evaluation also done in binned space?

medium: How large are tirex, moirai, and toto in parameter count?

medium: Figure A.9: How does performance vary with increasing prediction length? Peformance breakdowns in the appendices by short / medium / long forecasting would be quite helpful.

presentation: Gift-eval is not all univariate forecasting (see Table 1 in appendix) so I don’t think calling these experiments all univariate forecasting is totally accurate.

low/medium: Performance qualitatively looks much better for time series with strong periodicity but looks a bit strange for the more noisy forecasts (ie m4_quarterly/short , jena_weather, electricity/h/long). Do other time series forecasting models have these types of failure modes?

medium (but presentation): More description of the TabPFNv2 architecture would be helpful. My understanding is that we can roughly treat it as a blackbox, as it basically takes any input feature dimensionality and any number of output responses. However, clearly pointing this out would make the methodological description more clear – the authors use a fully pretrained model and their advance is really in just the feature construction techniques.

medium: I think there are probably missing comparisons to deep learning based, non-foundation models (ie N-hits (https://arxiv.org/abs/2201.12886) , FreTs (https://proceedings.neurips.cc/paper/2023/hash/f1d16af76939f476b5f040fd1398c0a3-Abstract.html)  and pathformer (https://arxiv.org/abs/2402.05956) to name a few)

medium: Have the authors tried finetuning TabPFNv2 on some of the datasets (say a subset of the tasks in the benchmarks)?

high: Does TabPFNv2 natively include some time series tasks in its pretraining? (I think the answer is yes – if so, any overlap in tasks evaluated here should be removed entirely).

medium: Did the authors try simply linearly detrending the response variable on the training set, and then using that to extrapolate to the test set? This would natively resolve the linear extrapolation problems.

High: Is Algorithm 1 applied to the whole response {y, y’} or just {y}, with the periodicity feature estimator extrapolated to {y, y’}?
Did the authors try other feature encodings beyond just cos/sin ones (ie EMAs of varying halflives)? Other feature embeddings would probably help on the more noisy sets of tasks.

---

> ### Author Response · Authors · 2026-04-14
> **Author's Response to Reviewer wWzS (part 1 out of 2)**
>
> Thank you for the thorough review. We address your comments below in order of priority. Due to the 5000-character limit, this response is split across two comments.
>
> **----- [High] -----**
>
> >The architecture is almost analogous to a MLP mixer style architecture in the sequence modeling domain. With that comes the tradeoff that we must know in advance how far to forecast, which is a pretty sizable limitation.
>
> We respectfully disagree. TabPFN-TS does not require a fixed horizon — like Chronos and TimesFM, it handles arbitrary prediction lengths by simply constructing X_test with the desired future timesteps. MLP-mixer models like TSMixer require the horizon to be fixed at training time, which is fundamentally different.
>
> > There's no evaluation of the uncertainty quantification beyond qualitatively of the model. One of the advantages of TabPFN is that it is natively probabilistic. Is the expected calibration error (or regression equivalents) actually useful?
>
> Thank you for the suggestion. We are running evaluations for Expected Calibration Error and will update the paper in two weeks as we need to rerun the baselines (their predictions are not available in GIFT-Eval but only their metrics).
>
> > How are the choices of bins for the binned predictive distribution constructed? Is the MSE evaluation also done in binned space?
>
> The bin grid is constructed at inference time from the range of observed training targets y_train. TabPFNv2 internally places a fixed-resolution discretization over this range and assigns probability mass to each bin. Point predictions and quantiles are derived from this distribution: the mean is the expectation over bin centers, and quantiles are read from the CDF. MASE and WQL are computed directly on the original target space and not in bin space.
>
> > why can the model not extrapolate a linear trend? I thought that the preprocessing already subtracts off the linear trend?
>
> The detrending in Algorithm 1 is applied only to improve spectral peak detection, not to the target values (y_train) passed to TabPFNv2. The model therefore receives raw, potentially trended targets. This limitation is structural and stems from TabPFNv2's interpolation-oriented pretraining objective (Section 6.1). We also experimented with explicit detrending but decided ultimately to remove it given that we saw little empirical improvement overall.
>
> > Does TabPFNv2 natively include some time series tasks in its pretraining?
>
> No. TabPFNv2 is pretrained exclusively on synthetically generated tabular datasets with no temporal structure, sequential ordering, or time-series-specific objective. There is therefore no overlap with any GIFT-Eval or fev-bench tasks. This is in fact the central point of the paper: the forecasting performance emerges from a model that has never seen time series data. Also, unlike other methods which use time series data for pretraining, we can be sure that no contamination occurred.
>
> >Is Algorithm 1 applied to the whole response {y, y’} or just {y}, with the periodicity feature estimator extrapolated to {y, y’}?
>
> Algorithm 1 is applied only to the observed historical targets y. The detected periodicities are then used to construct sinusoidal features for all timesteps (both historical and future) by evaluating the detected frequencies at the corresponding time indices.
>
> > Did the authors try other feature encodings beyond just cos/sin ones (ie EMAs of varying halflives)? Other feature embeddings would probably help on the more noisy sets of tasks.
>
> EMA-style features are inherently autoregressive (they require past target values to compute future values) which conflicts with our non-autoregressive multi-step formulation.
>
> > Why is TabPFNv2 so much slower at forecasting than models that are 100x its size?
>
> Two main factors: (1) TabPFN-TS requires a separate forward pass per series, whereas TSFMs amortize cost over large batches. (2) Most TSFMs apply patching, reducing effective sequence length ~16×. Introducing patching would require pretraining on time series data, contradicting the zero-shot premise. See Section 6.2 and Appendix A.10.
>
> *-- continued in the next comment --*

---

> > ### Comment · Reviewer_wWzS · 2026-05-02
> > **clarification**
> >
> > > We respectfully disagree. TabPFN-TS does not require a fixed horizon — like Chronos and TimesFM, it handles arbitrary prediction lengths by simply constructing X_test with the desired future timesteps. MLP-mixer models like TSMixer require the horizon to be fixed at training time, which is fundamentally different.
> >
> >
> > Thank you for the clarifications overall and especially on this point.
> >
> > > Introducing patching would require pretraining on time series data, contradicting the zero-shot premise.
> >
> > I am not sure this "matters" per se, other than your tabular foundation model being a reasonably good zero-shot predictor. But this ought to be less good than a time series foundation model being a zero-shot predictor of time series.

---

> ### Author Response · Authors · 2026-04-14
> **Author's Response to Reviewer wWzS (part 2 out of 2)**
>
> *-- second part of the response (due to character limit) --*
>
> **----- [Medium] -----**
>
> > How large are TiRex, Moirai, and Toto in parameter count?
>
> Moirai-2 (\~11.4M), TiRex (\~35M), Toto (\~151M).
>
> >  I think there are probably missing comparisons to deep learning based, non-foundation models (ie N-hits, FreTs and pathformer to name a few)
>
> Our baseline selection followed the GIFT-Eval leaderboard, one of the most comprehensive benchmarks available. DeepAR, TFT, and PatchTST are included as the most competitive deep learning baselines according to GIFT-Eval.
>
> > Have the authors tried finetuning TabPFNv2 on some of the datasets (say a subset of the tasks in the benchmarks)?
>
> We have explored this. Finetuning TabPFNv2 proved technically challenging and the resulting improvements were highly unstable. We consider these experiments inconclusive.
>
> > Did the authors try simply linearly detrending the response variable on the training set, and then using that to extrapolate to the test set? This would natively resolve the linear extrapolation problems.
>
> Yes. It helps significantly for series with clear sustained trends, but introduces errors when trends are noisy, nonlinear, or absent, making the improvement highly dataset-dependent. Empirically, we observed very small improvements and decided to mention detrending as an optional practitioner preprocessing step.

---

### Decision · Action_Editor_2b2Q · 2026-05-20

**Recommendation:** Accept with minor revision

**Additional Comments:**

Final Decision: Accept with minor revision.

All of the reviewers supported acceptance of the paper, hence I accept the paper. However, I ask for a minor revision, that would incorporate all changes requested by the reviewers. Especially include the promised to Reviewer wWzS evaluation, which is not included in the present revision:
"We are running evaluations for Expected Calibration Error and will update the paper in two weeks as we need to rerun the baselines (their predictions are not available in GIFT-Eval but only their metrics)."

**Audience:**

Yes

**Audience Explanation:**

The paper is of interest to the TMLR audience, especially researchers working on time-series modeling as well as tabular deep learning.

**Claims And Evidence:**

Yes

**Claims Explanation:**

Evaluation against the SOTA time-series foundation models on the fev-bench and gift-eval leaderboards supports the main findings of the paper. The empirical results support the claims made, regarding competitive univariate forecasting results and superior performance under covariate-based forecasting. Concerns raised in the review, particularly by adding stronger tabular baselines and clarifying the role of the TabPFN-v2 backbone versus temporal feature design were addressed.

---

> ### Author Response · Authors · 2026-06-22
> **Camera-Ready Version with Requested Changes**
>
> Dear Action Editor,
>
> Thank you for the decision and for tracking the remaining commitment.
> As promised to Reviewer wWzS, we have now added the Average Calibration Error evaluation (see Appendix 10 - Probabilistic Calibration Analysis), with a pointer aded from the main results in Section~4.1. The results show that TabPFN-TS’s calibration is comparable to most existing time-series foundation models, though not as good as TiRex.
>
> All other changes requested by the reviewers were already incorporated in the revision submitted prior to the AE's decision. We believe the paper is now complete with respect to all reviewer and AE requests.
>
> Thanks again!